# Treatment Responder Classification with Abstention

**Haoxiang Wang** [1 2]  **Aoqi Zuo** [3]  **Ziyan Wang** [4]  **Zhiheng Zhang** [4]
**Erdun Gao** [5 6]  **Kun Zhang** [7 2]  **Haoxuan Li**[✉ 1]  **Mingming Gong**[✉ 8 2]

## Abstract

Treatment responder classification seeks to learn a rule to classify individuals who will benefit from the treatment. This paper studies a new scenario in treatment responder classification when abstention is allowed, i.e., practitioners can opt out of making uncertain classification on some individuals for further investigation. By revealing the implicit relation between causal misclassification risk with abstention and Conditional Value at Risk (CVaR), we develop a doubly robust method named **TRECA** to learn the classification rule under loose convergence conditions on nuisance parameters, and further extend it to deal with possible violation on key assumptions such as monotonicity and unconfoundedness. Rigorous theories and extensive experiments on two real-world datasets demonstrate the theoretical and experimental guarantee on our methods in learning treatment responders classification rules with low regret at the cost of limited abstention.

## 1. Introduction

In a variety of fields where personalized decision-making are of interest, a key problem is to learn individual treatment rules to determine which individuals should be assigned treatments based on individual-level causal effects. To incorporate the population heterogeneity for more individualized and customized decision-making procedure, the target is to learn a function from a rich collection of individual covariates that predicts which treatment should be assigned to the individual. Comparing to classical decision-making tasks,

a key challenge in learning individual treatment rule is that we can only observe the factual outcomes corresponding to the treatments are actually assigned. On the other hand, the counterfactual outcome can never be observed, making the treatment effects especially those on individual level hard to identify in practice.

Treatment responder classification aims at learning a classification rule on treatment responders, i.e. individuals with a positive effect from the intervention of interest (Kallus, 2019; 2023). The necessity of classifying treatment responders has been widely addressed in applications (Atkinson et al., 2019; Gloster et al., 2020). For example, in personalized medicine, doctors want to prescribe drugs to individuals who will recover only with the drug taken. In recommendation system, practitioners want to push advertisement to the customers that would purchase the product had they been exposed to the advertisement, and would not purchase had the advertisement not been pushed. While existing frameworks on treatment responder classification even learning individual treatment rules require a deterministic decision be made for each individual, this can lead to significant error on some samples with high uncertainty. For this sake, decision-makers want the rule to opt out of making a decision when the confidence in classification is insufficient, to benefit more precise decision-making on remaining individuals with deterministic classification provided. We refer to such action as **abstention**, i.e. the rule abstains or rejects making decision for some individuals for further investigation. We provide the following example to illustrate the motivation of abstention, with an additional example provided in Appendix A.

**Clinical decision-making in existence of diagnostic ambiguity.** In clinical applications such as chronic and brain-related diseases, doctors want to prescribe effective but risky treatment such as invasive surgery to patients whose disease status is bound to benefit from such surgery, i.e., be a treatment responder. While it is not hard to classify treatment responder for some patients with good physical condition or obvious risk factors, doctors may observe rare lesion for some patients, and the uncertainty on these patients to benefit from the surgery may be significantly higher than others since there has been few samples with such lesion. Abstention learning, in this case, enables practitioners to

[1]Peking University [2]Mohamed bin Zayed University of Artificial Intelligence [3]The University of Sydney [4]Shanghai University of Finance and Economics [5]Australian Institute for Machine Learning [6]Adelaide University [7]Carnegie Mellon University [8]The University of Melbourne. Correspondence to: Haoxuan Li <hxli@stu.pku.edu.cn>, Mingming Gong <mingming.gong@unimelb.edu.au>.

*Proceedings of the 43$^{rd}$ International Conference on Machine Learning*, Seoul, South Korea. PMLR 306, 2026. Copyright 2026 by the author(s).

select these patients for further actions such as resorting to external information. By withholding judgment on these ambiguous cases for further information, abstention improves the overall accuracy of decisions for patients while saving the cost of expensive clinical testing, enhancing both patient safety and the efficiency of surgical planning.

While abstention learning has been studied in prediction and classification tasks which can be traced back to (Chow, 1957; 1970), it is not straightforward to extend to the causal scenarios as it faces the following key challenges. First, since treatment responder is determined by both factual and counterfactual outcomes, the identification of the loss and uncertainty terms requires extra assumption and derivation. Second, the identification of loss function in causal setting involves a series of nuisance parameters, which may affect the accuracy of learned treatment rule. This motivates us to develop robust learning methods to deal with possible mis-specification on nuisance parameters. However, prior robust methods in individual treatment rule learning or causal inference did not consider abstention as an option, while existing abstention learning methods also lack theoretical analysis in a causal context. Moreover, many existing works focus on cost-based abstention, which makes abstention based on a predefined cost of rejection that may be hard to interpret in practice (Cortes et al., 2016; Mao et al., 2024). Other works such as Herbei & Wegkamp (2006) study interval-based abstention to control the prevalence of rejection option based on the width of interval. However, these methods do not explicitly control the rejection rate, i.e., proportion of samples being abstained, which is a key feature to interpret the abstention strategy. More discussions on related works are provided in Appendix A.

Being aware of the challenges above, we develop a framework to classify treatment responders with abstention. The framework mainly consists of a predictor for classification and a rejector to abstain samples. We start by formulating the misclassification risk using potential outcomes, and provide a constraint-based definition on the targeted abstention rule for enhanced interpretability in application. By revealing the relation between causal misclassification risk with abstention and Conditional Value at Risk (CVaR), a value widely studied in economics and leveraging the properties of CVaR, we develop a doubly robust method **T**reatment **RE**sponder **C**lassification with **A**bstention (**TRECA**) to learn the classification rule through identification and estimation on the loss function. Taking account of possible violation in key assumptions such as monotonicity and unconfoundedness, we propose modification on the method to learn the predictor through minimizing tight upper bound derived on the loss function under partial identification. Comparing with previous abstention learning methods, we formulate the problem by making explicit constraint on the expected rejection rate, which enhances the interpretation

of the method. Supportive theories have been developed to guarantee the convergence on loss function under loose convergence of nuisance parameters and the accuracy of learned rejector, with extensive experiments on two real-world datasets demonstrate the performance of our methods in a variety of real-world scenarios. Our main contributions are summarized as follows:

- To the best of our knowledge, this is the first work considering causal decision making when abstention is optional, greatly extending the application of treatment responder classification. It also reveals the relation between abstention and CVaR, which can serve as a springboard to inspire further studies on other causal decision making methods with abstention.

- The paper proposes a comprehensive framework to classify treatment responder in various scenarios with and without monotonicity assumption, derives improved partial identification bounds, and proposes robust estimators on the loss function with proper theoretical guarantees.

- Extensive experiments on real-world datasets demonstrate the superiority of our method in classifying treatment responders under varying abstention rates.

## 2. Problem Setup

We start by introducing the basic setup as well as key notations throughout the paper, with a summary table provided in Appendix B. Consider a group of $n$ units sampled from superpopulation $\mathcal{P}$. Each unit with covariates $X \in \mathcal{X}$ where $\mathcal{X}$ is a bounded vector space receives a binary treatment $T \in \{-1, 1\}$ and produce a binary outcome $Y \in \{-1, 1\}$. Let $Y(1)$ and $Y(-1)$ be the potential outcome under treatment $T = 1$ and $T = -1$, and $Y$ be the observed outcome. In practice, $Y$ can be a binary categorization from a continuous outcome indicating whether the outcome is favored. The individuals are divided into two classes: treatment responders with $Y(1) > Y(-1)$ and non-responders with $Y(1) \leq Y(-1)$. For notation simplicity, let $R = 2I(Y(1) > Y(-1)) - 1$ indicate whether an individual is a treatment responder with $R = 1$ or non-responder with $R = -1$. Let $\rho(X) = \mathbb{P}(R = 1 \mid X)$ be the conditional responding probability, and $\tau(x) = \mathbb{E}[Y(1) - Y(-1) \mid X = x]$ be the conditional average treatment effect (CATE). Denote the propensity score $\pi(X) = \mathbb{P}(T = 1|X)$ and conditional outcome expectation $\mu_t(x) = \mathbb{E}[Y \mid T = t, X = x]$, $\mu(x) = (\mu_1(x), \mu_{-1}(x))$.

Our framework consists of learning a predictor $f : \mathcal{X} \to \{-1, 1\}$ that predicts whether an individual is a treatment responder when $f(x) = 1$ or non-responder when $f(x) = -1$ based on the covariates, as well as an abstention rule $r : \mathcal{X} \to \{0, 1\}$ which abstains making decision on the sample with $X = x$ when $r(x) = 1$ while retain the sample

with $r(x) = 0$. Let $\mathcal{F}$ and $\mathcal{R}$ be the hypothesis set of $f$ and $r$ respectively. For $\delta \in (0,1)$, our goal is to the $\delta$-optimal rule defined as follows

$$(f^*, r^*) = \arg \min_{(f,r) \in \mathcal{F} \times \mathcal{R}} \tilde{L}_\theta(f,r) \text{ s.t. } \mathbb{E}\{r(X)\} \leq \delta, \quad (1)$$

where $\tilde{L}_\theta(f,r) = \theta \cdot \mathbb{P}(f(X) = 1, R = -1, r(X) = 0) + (1-\theta) \cdot \mathbb{P}(f(X) = -1, R = 1, r(X) = 0)$. Term $\tilde{L}_\theta(f,r)$ is the misclassification loss predicting $R$ from $X$ on samples that have not been abstained. Hyper-parameter $\theta$ weights the importance between false positive and false negative samples to match the requirement in practice. For example, in the classification on drug responders, doctors may prefer false negative to false positive samples for safety concern. The condition $\mathbb{E}\{r(X)\} \leq \delta$ ensures that the abstention rule abstain less than $\delta$-proportion of samples on the superpopulation $\mathcal{P}$ to ensure its generalization performance. Smaller $\delta$ implies that less individuals are expected to be abstained, which generally leads to an increased risk to retained samples. Therefore, the choice of $\delta$ relies on a risk-coverage trade-off taken account by practitioners in application. Comparing to the cost-based abstention where the choice of abstention is assumed to induce a pre-specified cost (Cortes et al., 2016; Mao et al., 2024), the constraint-based abstention rule that determines the proportion of abstained units avoids explicit definition on the reject cost, thus is more interpretable in many application scenarios (Franc et al., 2023). In the special case when $\delta = 0$, i.e., no samples being abstained, $\tilde{L}_\theta(f,r)$ degenerates to the misclassification loss discussed in previous works on treatment responder classification such as Kallus (2019); Wu et al. (2025). Our basic framework is built upon the following classical assumptions in causal inference:

**Assumption 2.1** (Consistency). The observed outcome equals to the potential outcome under assigned treatment, i.e. $[Y_i \mid T_i = t] = Y_i(t)$ for any unit and $t \in \mathcal{T}$.

**Assumption 2.2** (Positivity). There exists $0 < \underline{\pi} < 1$ such that $\forall X \in \mathcal{X}$, $\pi(x) \in [\underline{\pi}, 1 - \underline{\pi}]$.

**Assumption 2.3** (Unconfoundedness). $(Y(0), Y(1)) \perp\!\!\!\perp T|X$.

Assumptions 2.1-2.3 are standard assumptions made in causal inference, specifically in the task of treatment responder classification (Kallus, 2019). Despite of this, in Section 3.3, we also discuss the extension of our method in violation of unconfoundedness assumption. Intuitively, a good abstention rule that achieves low misclassification loss with limited abstention is an **uncertainty-based** criterion, which rejects samples that are highly uncertain to ensure better classification accuracy on retained samples. In the following discussion, we will provide a theoretically-grounded measurement on the uncertainty in treatment responder classification task using conditional risk and derive its identification under rigorous conditions.

## 2.1. Misclassification Loss under Abstention is Conditional Value at Risk

To start with, we introduce the following conditional risk for uncertainty measurement[1]

$$V_f(x) = \mathbb{E}\{\theta I(f(X) = 1, R = -1) + (1-\theta)I(f(X) = -1, R = 1) \mid X = x\}. \quad (2)$$

The conditional risk measures the uncertainty on $f$ predicting the true responder status $R$ given covariates. Due to the randomness at individual level, $R$ is not fixed given covariates. Such randomness is also the key difference between individual treatment effect (ITE) and conditional average treatment effect (CATE) (Vegetabile, 2021). Therefore, the conditional expectation taken in (2) originates from the individual level, and high conditional risk indicates high uncertainty on the responder type among individuals given the same covariate observation. In remark, conditional risk is different from the Conditional Value at Risk discussed later. Denote $F_{V_f}$ as the cumulative probability function of $V_f(X)$, and for $\alpha \in (0,1)$, define $F_{V_f}^{-1}(\alpha) = \inf\{v \mid F_{V_f}(v) \geq \alpha\}$. We impose the following minor regularity assumptions on the conditional risk $V_f(X)$:

**Assumption 2.4** (Local Regularity). The function $F_{V_f}$ is continuously differentiable at $F_{V_f}^{-1}(1-\delta)$ with positive derivative, i.e., $F'_{V_f}(F_{V_f}^{-1}(1-\delta)) > 0$.

Assumption 2.4 imposes regularity assumption on the cumulative distribution function in the locality of $(1-\delta)$-quantile. The positivity of derivative assumes that the quantile $F_{V_f}^{-1}(1-\delta)$ is well defined to ensure uniqueness on the optimal abstention rule. The continuous differentiability condition is applied to derive robustness property on the loss function.

**Theorem 2.5.** *Under Assumptions 2.1-2.4, the $\delta$-optimal rule has the following equivalent form:*

$$f^* = \arg \min_{f \in \mathcal{F}} L_{\theta,\delta}(f)$$

*with*

$$L_{\theta,\delta}(f) = \mathbb{E}\{\theta I(f(X) = 1, R = -1) + (1-\theta)I(f(X) = -1, R = 1)|V_f(X) \leq F_{V_f}^{-1}(1-\delta)\}, \quad (3)$$

*and $r^*(x) = I(V_{f^*}(X) > F_{V_{f^*}}^{-1}(1-\delta))$. Moreover, under Assumption 2.1 and 2.3, we have*

$$L_{\theta,\delta}(f) = \mathbb{E}\left\{V_f(x) \mid V_f(x) \leq F_{V_f}^{-1}(1-\delta)\right\}. \quad (4)$$

Theorem 2.5 indicates that the optimal predictor that minimizes the misclassification loss $\tilde{L}_\theta(f,r)$ on retained samples under constraint on abstention rate is equivalent to the

---

[1]For the notation on conditional risk, we omit $\theta$ in default for simplicity.

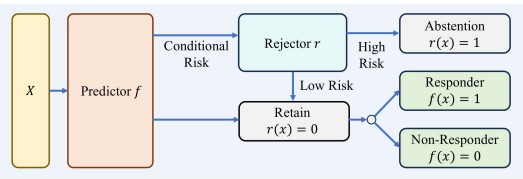

*Figure 1.* Framework on treatment responder classification with abstention.

minimizer of $L_{\theta,\delta}(f)$ defined in (3). Given the optimal predictor $f^*$, the optimal abstention rule abstains samples by thresholding the $(100\delta)\%$ highest conditional risk. This inspires us to train the predictor and rejector in a sequential order. In the first step, we train the predictor through minimizing the empirical form of $L_{\theta,\delta}(f)$ and estimate the conditional risk for each individual. The abstention rule is thereon determined as a thresholding rule rejecting samples with the highest conditional risk for better accuracy on the retained samples with relatively low conditional risk. A diagram on the overall framework is presented in Figure 1.

The loss function in (4) represents the expectation on the conditional risk $V_f(x)$ among the $(100\delta)\%$ lowest values. Such average among subpopulation with the lowest scores has been explored as the Conditional Value at Risk (CVaR). CVaR commonly appears to be the target of interest in financial engineering or management studies, where the extreme behaviors are of interest (Krokhmal et al., 2002; Sarykalin et al., 2008; Filippi et al., 2020). While multiple works discussed about the optimization, estimation and asymptotic properties on CVaR (Uryasev, 2000; Rockafellar et al., 2000; Chun et al., 2012), in particular, Kallus (2023) developed the statistical inference procedure on CVaR in terms of ITE, studying CVaR in the context of individual treatment rule learning remains little explored. Inspired by Rockafellar et al. (2000), we have the following equivalent form on loss function, which will be leveraged to construct estimators in the next section.

**Proposition 2.6.** *Under Assumptions 2.1-2.3, the loss function $L_{\theta,\delta}(f)$ defined in (3) has the following equivalent form:*

$$L_{\theta,\delta}(f) = \sup_{\beta \in \mathbb{R}} \left\{ \beta + \frac{1}{1-\delta} E(V_f(X) - \beta)_- \right\},$$

*where $(u)_- = u \wedge 0$, $r^*(x) = I(V_f(X) > \beta^*)$, and $\beta^* = \arg\max_{\beta \in \mathbb{R}} \left\{ \beta + \frac{1}{1-\delta} \mathbb{E}(V_f(X) - \beta)_- \right\}$.*

Proposition 2.6 formulates the loss function as the supremum over an additional parameter $\beta$. The optimal value $\beta^*$ coincides with the threshold for the abstention rule. In practice, we can either rank the estimates $\{\hat{V}_f(X_i)\}_{i=1}^n$ and abstain the highest $(100\delta)\%$ values to ensure a strict abstention rate $\delta$ or use the optimizer $\hat{\beta}$ as the threshold to enhance efficiency and generalizability. The details for loss

estimation and optimization will be discussed in the next section.

## 3. Methodology and Theories

### 3.1. Identification and Estimation of Loss Function under Monotonicity

From Proposition 2.6, minimizing the loss function in (3) is estimable once the conditional risk $V_f(x)$ is identified. In light of this, we develop the identification on loss function $L_{\theta,\delta}(f)$ through leveraging the following monotonicity assumption also made in Kallus (2019):

**Assumption 3.1** (Monotonicity). $Y(1) \geq Y(-1)$.

As shown in Table 1, monotonicity assumption essentially implies that the subgroup with $Y(1) = -1, Y(-1) = 1$ do not exist. In drug-treatment example, it means that the drug will not have a harmful effect on the patient. While it is commonly made in causal effect identification such as distributional treatment effect (Kim, 2014) and local average treatment effect (Angrist & Imbens, 1995), there are cases that monotonicity does not hold. In Section 3.2 discuss the extension of our method when monotonicity assumption may not hold. The conditional risk is identified by the following proposition:

**Proposition 3.2.** *Under Assumptions 2.1-2.3 and 3.1, $V_f(x) = \frac{1}{2}f(x)\{\theta - \rho(x)\} + \frac{1}{2}\{\theta + (1 - 2\theta)\rho(x)\}$, and $\rho(x) = \tau(x)/2 + P(Y(1) < Y(-1) \mid X = x)$. When Assumption 3.1 holds, $\rho(x) = \tau(x)/2$.*

*Table 1.* Categorization of $(Y(1), Y(-1))$ when monotonicity holds or violates.

| Monotonicity | Hold (✓) | May not hold (?) |
|---|---|---|
| Responder | $(+1, -1)$ | $(+1, -1), (-1, +1)$ |
| Non-responder | $(+1, +1), (-1, -1)$ | $(+1, +1), (-1, -1)$ |

We write $V_f(x; \tau)$ substituting $\rho(x)$ in $V_f(x)$ with $\tau(x)/2$ to address that the identification involves $\tau$. The identification in Proposition 3.2 introduces CATE $\tau(x)$ as nuisance parameter that affects the performance of targeted estimator. The concern on nuisance parameters has also been aroused in a wide range of causal studies, including effect estimation, policy learning and causal machine learning (Bang & Robins, 2005; Chernozhukov et al., 2018; Moosavi et al., 2023). A straightforward way to optimize $L_{\theta,\delta}(f)$ in practice is to minimize its naive empirical estimator $\hat{L}_{\theta,\delta}^{naive}(f; \hat{\tau}) = \sup_{\beta \in \mathbb{R}} \left\{ \beta + \frac{1}{n(1-\delta)} \sum_{i=1}^n (V_f(X_i; \hat{\tau}) - \beta)_- \right\}$, where $V_f(x; \hat{\tau})$ is the plug-in estimator of $V_f(x; \tau)$. However, as the loss is sensitive to the estimation on $\tau$ as well as $\beta$, when $\hat{\tau}$ or $\hat{\beta}$ is non-negligibly biased or converges slowly, $\hat{L}_{\theta,\delta}^{naive}(f; \hat{\tau})$ may not convergence to $L_{\theta,\delta}(f)$ at a decent rate. Taking account of this, we construct a robust

estimation on loss function that is less sensitive to the accuracy of $\hat{\tau}(x)$ as follows. The augmented loss function $l_{\theta,\delta}$ is constructed by substituting CATE $\tau(X)$ in $V_f(X;\tau) - \beta$ with a doubly robust estimator modeled by $\mu$ and $\pi$ (Bang & Robins, 2005; Chernozhukov et al., 2018):

$$
\hat{L}_{\theta,\delta}(f;\hat{\tau},\hat{\mu},\hat{\pi})
$$
$$
= \sup_{\beta\in\mathbb{R}} \left\{ \frac{1}{n}\sum_{i=1}^{n} l_{\theta,\delta}(X_i,T_i,Y_i,f;\hat{\tau},\hat{\mu},\hat{\pi},\beta) \right\}, \quad (5)
$$

where

$$
l_{\theta,\delta}(X,T,Y,f;\tau,\mu,\pi,\beta)
$$
$$
= \beta + \frac{1}{1-\delta} I(V_f(X;\tau)\le\beta)\Big[\frac{1}{2}(f(X)+1)\theta
$$
$$
+ \frac{1}{4}\{1-2\theta-f(X)\}\times\Big\{\mu_1(X)-\mu_{-1}(X)
$$
$$
+ \frac{(T+1)/2-\pi(X)}{\pi(X)(1-\pi(X))}(Y-\mu_T(X))\Big\} - \beta\Big].
$$

**Training algorithm.** From (5), minimizing the loss function $\hat{L}_{\theta,\delta}(f;\hat{\tau},\hat{\mu},\hat{\pi})$ is a mini-max problem involving nuisance parameters $(\tau,\mu,\pi,\beta)$. We propose an iterative algorithm **TRECA** to learn the predictor $f$ and nuisance parameters iteratively. In each iteration, parameter $\beta$ is updated through gradient ascent as to maximize the empirical mean of $l_{\theta,\delta}$ in (5), while predictor $f$ and other parameters $(\tau,\mu,\pi)$ are updated through gradient descent. The details are presented in Algorithm 1.

Recall that $\beta^* = \arg\max_{\beta\in\mathbb{R}} l_{\theta,\delta}\mathbb{E}(X,T,Y,f;\tau,\mu,\pi,\beta)$, with $(\tau,\mu,\pi)$ be the true functions. Comparing with the naive estimator $\hat{L}_{\theta,\delta}^{naive}(f;\hat{\tau})$, $\mathbb{E}l_{\theta,\delta}(X,T,Y,f;\hat{\tau},\hat{\mu},\hat{\pi},\hat{\beta})$ has zero partial derivative at $(\tau,\mu,\pi,\beta^*)$, which also known as Neyman orthogonality condition. This condition ensures that the expectation on $l_{\theta,\delta}$ as well as its empirical mean is not sensitive to small error on nuisance parameters near the true value, thus ensuing double robustness property (Kallus, 2023), which is formally stated by the following theorem:

**Theorem 3.3.** *Suppose Assumptions 2.1-2.3 and 3.1 hold, and there exists $\tilde{\pi}\in[\underline{\pi},1-\underline{\pi}]$ such that the estimators for nuisance parameters satisfy $\|\hat{\pi}-\tilde{\pi}\| = o_p(1)$, $\|\hat{\mu}-\tilde{\mu}\| = o_p(1)$, and $\|\hat{\tau}-\tau\|_q = O_p(\epsilon_n^{\frac{q}{q+1}})$ for some $q>1$. Then when either of the following conditions holds:*

$$
\|\hat{\pi}-\pi\| = o_p(\epsilon_n) \qquad \|\hat{\mu}-\mu\| = o_p(\epsilon_n),
$$

*we have $|\hat{L}_{\theta,\delta}(f;\hat{\tau},\hat{\mu},\hat{\pi}) - L_{\theta,\delta}(f)| = O_p(\epsilon_n \vee n^{-1/2})$.*

Theorem 3.3 suggests that even if either $\pi$ or $\mu$ is inconsistent, we can get a consistent estimator on loss function once we have a mild consistent estimator on $\tau$. Note that while $\tau(x) = \mu_1(x) - \mu_{-1}(x)$, $\mu$ and $\tau$ are treated as separated parameters which means that they do not require

the same model. For instance, $\hat{\tau}$ can be estimated through direct CATE models without modeling the conditional expectations such as causal forests (Wager & Athey, 2018) or R-learner (Nie & Wager, 2021). The proof on Theorem 3.3 is provided in the Appendix. Note that from the proof details, the error bound is linear to the constants $c$, $(1-\delta)^{-1}$ and $\underline{\pi}^{-1}$ involved in the proof. Therefore, the convergence rate is not sensitive to the problem-specific constants. We also derive generalization bound on $\hat{L}_{\theta,\delta}(f;\hat{\tau},\hat{\mu},\hat{\pi})$ by measuring space complexity through Rademacher complexity, see Theorem D.2 in appendix for details.

### 3.2. Discussion When Monotonicity Does Not Hold

As mentioned before, although monotonicity assumption is commonly made in identification of causal parameters, there are real-world scenarios where it may violate. In this section, we consider the case in which $V_f(x)$ as well as loss function $L_{\theta,\delta}(f)$ can not be fully identified when monotonicity assumption may not hold. Instead, we seek to minimize an upper bound on loss function. Recall the expression and definition on $\beta^*$ in Proposition 2.6, if we can find a tight upper bound $U_f(x)$ on conditional risk, i.e. $U_f(x) \ge V_f(x)$ almost everywhere, then $M_{\theta,\delta}(f) := \sup_{\beta\in\mathbb{R}} \left\{ \beta + \frac{1}{1-\delta}\mathbb{E}(U_f(x)-\beta)_- \right\}$ is a tight upper bound on $L_{\theta,\delta}(f)$ since $M_{\theta,\delta}(f) \ge \beta^* + \frac{1}{1-\delta}\mathbb{E}(U_f(x)-\beta^*)_- \ge \beta^* + \frac{1}{1-\delta}\mathbb{E}(V_f(X)-\beta^*)_- = L_{\theta,\delta}(f)$.

**Proposition 3.4.** *For functions $\rho^L(x), \rho^U(x)$ such that $\rho(x) \in (\rho^L(x), \rho^U(x))$ almost everywhere, we have the following tight upper bound on conditional risk*

$$
V_f(x) \le \frac{1-f(x)}{2}\cdot(1-\theta)\rho^U(x) + \frac{1+f(x)}{2}\cdot\theta(1-\rho^L(x)). \quad (6)
$$

Proposition 3.4 shows that to construct upper bound on conditional risk $V_f(x)$, it suffices to derive bounds to partially identify $\rho(x)$. The proposed bound (6) improves Theorem 4.4 in (Wu et al., 2025) in the sense that given $f(x) \in \{-1,1\}$, equality in (6) is attained once $\rho(x) = \rho^U(x)$ or $\rho(x) = \rho^L(x)$ but not necessarily both. From Proposition 3.2, $\rho(x) = \frac{\tau(x)}{2} + \mathbb{P}(Y(1) < Y(-1) \mid X = x)$. The later term $\mathbb{P}(Y(1) < Y(-1) \mid X = x)$, has been explored in previous works as the fraction negatively affected (FNA) (Kallus, 2022). In this paper, we adopt the upper and lower bounds in the form $\rho^U(x) = u(\tau(x))$ and $\rho^L(x) = w(\tau(x))$, where $u$ and $w$ are two known functions that are differentiable with regard to $\tau$. In practice we take the bound in Wu et al. (2025) with $w(\tau) = \max\{\frac{\tau}{2},0\}$, and $u(\tau) = \frac{1}{2} + \frac{\tau}{4}$.

In conclusion, let $U_f(x;\tau)$ be the upper bound on conditional risk derived in (3.4). The debiased estimator for upper

bound on loss function $M_{\theta,\delta}(f)$ is constructed as

$$
\begin{aligned}
&\widehat{M}_{\theta,\delta}(f; \hat{\tau}, \hat{\mu}, \hat{\pi}) \\
&= \sup_{\beta \in \mathbb{R}} \left\{ \frac{1}{n} \sum_{i=1}^{n} m_{\theta,\delta}(X_i, T_i, Y_i, f; \hat{\tau}, \hat{\mu}, \hat{\pi}, \beta) \right\},
\end{aligned}
\tag{7}
$$

$$
\begin{aligned}
&m_{\theta,\delta}(X, T, Y, f; \tau, \mu, \pi, \beta) \\
&= \beta + \frac{1}{1-\delta} I(U_f(X;\tau) \leq \beta) \Big[ U_f(X;\tau) - \beta \\
&\quad + \partial_\tau U_f(X;\tau) \frac{(T+1)/2 - \pi(X)}{\pi(X)(1-\pi(X))} (Y - \mu_T(X)) \Big],
\end{aligned}
$$

where $\partial_\tau U_f(X;\tau) = \frac{(1-f(x))(1-\theta)}{2} u'(\tau(x)) - \frac{(1+f(x))\theta}{2} w'(\tau(x))$, and $w', u'$ are the first-order derivatives of $w, u$. Therefore, we can learn the predictor $f$ and nuisance parameters $(\hat{\tau}, \hat{\pi}, \beta)$ iteratively by replacing the loss function with its upper bound $m_{\theta,\delta}$. The detailed algorithm on the modified method is presented in Algorithm 2.

Note that $(u)_- = \mathbb{E}uI(u \leq 0)$, the augmented loss function $m_{\theta,\delta}$ appends a re-weighted doubly robust residual to the latter term $U_f(X;\tau) - \beta$. While such residual has zero mean when either of $\mu, \pi$ is specified, the double robustness property still holds by check the orthogonality of nuisance parameters $(\hat{\tau}, \hat{\beta})$ on $\mathbb{E}m_{\theta,\delta}(X, T, Y, f; \tau, \mu, \pi, \hat{\beta})$. The double robustness property is rigorously formulated in Theorem 3.5 as below.

**Theorem 3.5.** *Suppose Assumption 2.1-2.4 hold, and the cumulative distribution function of $U_f(x; \hat{\tau})$ is continuous differentiable with positive derivative at $\beta^*$. Then for estimators $(\hat{\tau}, \hat{\mu}, \hat{\pi})$ satisfying the same convergence and doubly robustness conditions as in Theorem 3.3, we have*

$$
|\hat{M}_{\theta,\delta}(f; \hat{\tau}, \hat{\mu}, \hat{\pi}) - M_{\theta,\delta}(f)| = O_p(\epsilon_n \vee n^{-1/2}).
$$

Finally, to provide further theoretical guarantee on the performance of abstention rule learned under partial identification, we develop Theorem 3.6 to bound the rate on misclassification probability of the abstention rule learned from the upper bound on loss function derived under partial identification $\rho(x) \in (\rho^L(x), \rho^U(x))$. Let $\hat{\rho}^U(x) = u(\hat{\tau}(x))$, $\hat{\rho}^L(x) = w(\hat{\tau}(x))$ be the estimators on sensitivity bound, and let $d_\infty = \max_{x \in \mathcal{X}} |\hat{\rho}^U(x) - \hat{\rho}^L(x)|$ denote the largest length over $x$. We have the following theorem:

**Theorem 3.6.** *Given estimators $(\hat{\tau}, \hat{\mu}, \hat{\pi})$ satisfying Assumption 2.1-2.4 and conditions in Theorem 3.3, let $\hat{f}_M = \arg\min_{f \in \mathcal{F}} \hat{M}_{\theta,\delta}(f; \hat{\tau}, \hat{\mu}, \hat{\pi})$ be the treatment rule learned from the upper bound on classification loss and $\hat{r}^M(x)$ be the ;earned abstention rule. Assume that (a) $\rho(X)$ has bounded density function, and (b) $\hat{f}_M(x) = -1$ for $\rho^U(x; \hat{\tau}) \leq \theta$ and $\hat{f}_M(x) = 1$ for $\rho^L(x; \hat{\tau}) > \theta$, we have*

$$
\mathbb{P}(\hat{r}_M(X) \neq r^*(X)) = O(d_\infty \vee n^{-1/2}).
$$

Theorem 3.6 shows that for sufficiently large hypothesis set, narrower the partial identification bound is, more accurate the learned rule will be. In particular, when $\rho(x)$ is fully identified, i.e., $d_\infty = 0$, we can learn the $\delta$-optimal abstention rule at an error rate of at most $n^{-1/2}$. Condition *(b)* assumes that the optimal predictor can classify the responder correctly for deterministic samples under partial identification. In particular, condition *(b)* is satisfied when the VC dimension of hypothesis set $\mathcal{F}$, $\text{VC}(\mathcal{F}) \geq n$ such that the sample points can be shattered by the models in $\mathcal{F}$. The hypothesis set $\mathcal{F}$ with sufficiently large VC dimension can be achieved through various machine learning models. For instance, for ReLU networks, it is shown that $\text{VC}(\mathcal{F}) \gtrsim WL \log(W/L)$, where $W$ and $L$ are the number of nodes and layers (Bartlett et al., 2019). Therefore, condition *(b)* is satisfied by estimating $f$ through ReLU network with sufficient amount of nodes and layers $WL = O(n)$.

### 3.3. Discussion When Unconfoundedness Does Not Hold

In this section, we extend the method in existence of unmeasured confounders. The key problem is that $\tau(x)$ can not be fully identified. We deal with it by making sensitivity analysis on CATE and derive bounds on conditional risk $V_f(x)$. Suppose the sensitivity bound constructed on $\tau(x)$ is $\tau(x) \in [\tau^L(x), \tau^U(x)]$. Then from Proposition 3.2, under Assumptions 2.1, 2.2 and 2.4, $\rho(x) \in [\tau^L(x)/2, \tau^U(x)/2]$. Therefore, from Proposition 3.4, we have $V_f(x) \leq U_f(X)$, with $U_f(X)$ defined as

$$
U_f(X) = (1-\theta)\frac{1-f(x)}{2}\frac{\tau^U(x)}{2} + \theta\frac{1+f(x)}{2}(1 - \frac{\tau^L(x)}{2})
$$

and hence from above we have $L_{\theta,\delta}(f) \leq \sup_{\beta \in \mathbb{R}} \Big\{ \beta + \frac{1}{1-\delta} \mathbb{E}(U_f(X) - \beta)_- \Big\}$. The treatment responder is selected through minimizing the upper bound above with the same procedure as in Algorithm 2. In practice, we adopt the plug-in estimator on the following sensitivity bound inspired by the sensitivity bound discussed in Section 22.3 in Imbens & Rubin (2015):

$$
\begin{aligned}
\tau^U(x) &= \pi(x)\mu_{1,1}(x) + 1 - [1 - \pi(x)]\mu_{-1,-1}(x); \\
\tau^L(x) &= \pi(x)\mu_{1,1}(x) - 1 - [1 - \pi(x)]\mu_{-1,-1}(x),
\end{aligned}
$$

where $\mu_{t,t'}(x) = \mathbb{E}(Y(t) \mid T = t', X = x)$. Experimental results in Section 4 demonstrate that the above modification of TRECA in relaxation of unconfoundedness assumption performs well comparing with baseline methods in making more accurate classification on treatment responders among retained samples.

### 3.4. Time and Space Complexity Analysis

**Time complexity.** Let $n$ be the sample size, $m$ the batch size, and let $l_\pi, l_\mu, l_\tau, l_f$ and $d_\pi, d_\mu, d_\tau, d_f$ denote the number of layers and hidden dimensions for the propensity,

outcome, CATE, and predictor networks, respectively. Let $t_\phi = l_\phi d_\phi^2$ represent the time complexity of a one-time forward pass in network $\phi$. The analysis of the time complexity of our method can be divided into three stages. In Stage 1, $\hat{\pi}(x)$ and $\hat{\mu}_t(x)$ are trained by minimizing the MSE over the entire samples, which results in a complexity of $O(n(t_\pi + 2t_\mu))$. In Stage 2, we train the CATE $\tau$ and predictor $f$ alternatively by minimizing the CVaR loss. Estimating the CVaR loss requires calling $\hat{\pi}(x)$ and $\hat{\mu}_t(x)$, which is performed $r_\beta$ rounds for each of the $n/m$ epochs, resulting in a time complexity of $O(nm^{-1}r_\beta(t_\pi + t_\mu))$. The CATE and predictor are then trained with complexity $O(n(t_\tau + t_f))$. Balancing the representations in CATE estimation involves a time complexity of $O(n/m \cdot m^2) = O(nm)$. In Stage 3, the conditional risks $V_{\hat{f}}(x)$ are computed for each individual using the trained predictor, with time complexity $O(n(t_\tau + t_f))$. In summary, the overall time complexity is $O(n[2(t_\tau + t_f) + (1 + m^{-1}r_\beta)(d_\pi + d_\mu) + m])$, which is linear to all related parameters.

**Space complexity.** Following the notations and stage division in time complexity, further define $s_\phi = d_\phi l_\phi$ to be the capacity of some network $\phi$. In Stage 1, the space complexity is equal to the network capacity times batch size $O(m(s_\pi + s_\mu))$. In Stage 2, since the model is trained through alternative gradient descent on $\tau$ and $f$ while requires calling the propensity for each individual, the space complexity for weight storage, gradient memory and propensity storage is $O(m(s_\tau + s_f))$. Computing the representation distance in learning CATE involves $O(m^2)$ space. Learning the abstention rule in Stage 3 requires $O(1)$ extra space. In summary, the space complexity is $O(m(s_\pi + s_\mu + s_\tau + s_f) + m^2)$, which is at most quadratic to relevant parameters. In Appendix E.4, experiments indicate that **TRECA** has a medium level of execution time and memory comparing with baseline methods, providing a balance between efficiency and accuracy.

# 4. Experimental Results

## 4.1. Experimental Setup

**Datasets and Preprocessing.** To measure the effectiveness of the proposed methods, we conduct extensive experiments on two real-world datasets, **Twins** (Almond et al., 2005) and **Jobs** (LaLonde, 1986) and their transformed datasets **Twins_mono** and **Jobs_mono** to ensure monotonicity. See Appendix E for supplementary details on datasets, preprocessing procedure and configurations, and see https://github.com/hxjimmywang/TRECA_ICML for code.

**Baselines.** We compare our results with the treatment rules derived from the CATE estimators learned by the following methods. (1) Meta-learners, which combines multiple models to enhance performance, such as **X-learner** and

**T-learner** (Künzel et al., 2019; Salditt et al., 2024). (2) Counterfactual Representation Learning (CFR) based models, including **CFRNet** (Shalit et al., 2017), **DeRCFR** (Wu et al., 2022), **ESCFR**(Wang et al., 2023), **DragonNet** (Shi et al., 2019) and **CFRISW** (Hassanpour & Greiner, 2019). (3) Other methods, including tree-based method Causal Forest **CF** (Athey & Imbens, 2016) and CAE-based method **CEVAE** (Louizos et al., 2017). We also compare our methods with **CTRL** (Wu et al., 2025) which deals with treatment responder classification without taking account of abstention. For fair comparison, we apply uncertainty-based abstention strategy for baseline methods which abstains samples with high conditional risk to ensure that results are measured on the same amount of retained samples, with details provided in Appendix E.2.

**Evaluation Metrics.** For a sample set $\mathcal{M}$ with size $m$, following Wu et al. (2025), we evaluate the accuracy in treatment responder classification through the regret on predictor $f$ **f-regret**, defined by **f-regret** $= \theta \cdot$ **FNR** $+ (1 - \theta) \cdot$ **FPR**, where **FNR** $= \sum_{i \in \mathcal{M}} I(\hat{f}(X_i) = -1, R_i = 1)/|\mathcal{M}|$, **FPR** $= \sum_{i \in \mathcal{M}} I(\hat{f}(X_i) = 1, R_i = -1)/|\mathcal{M}|$ are the false negative and positive rates of predicting the responder $R_i$. The metrics of interest are the **f-regret** computed among three subsets: **f-regret-r** on retained samples $\hat{r}(X_i) = 0$, **f-regret-a** on abstained samples $\hat{r}(X_i) = 1$ and **f-regret-o** on the overall dataset. We take $\theta = 0.5$ throughout the experiment, and evaluate the in-sample and out-of-sample performances by replicating experiment five rounds to report the mean and standard deviation.

**Model Structure.** In **TRECA** and its modifications, we use CFRNet (Shalit et al., 2017) in train the CATE estimator $\hat{\tau}$ and condition expectation $\hat{\mu}$. CFRNet is chosen for two main reasons. (1) Through balancing representation distributions in factual and counterfactual outcomes, experimental results shows that counterfactual representation learning (CFR) has good performance in classifying treatment responders. Previous work also recommends CFR-based methods to learn CATE in treatment responder classification task (Wu et al., 2025). (2) The bifurcated structure of CFRNet with each branch predicting factual and counterfactual outcomes respectively ensures good fitness on $\mu$, which is also theoretically guaranteed in Shalit et al. (2017) to ensure low generalization bound. We choose MLP with two hidden layers to learn the rejector and predictor. To derive gradient on loss function, we use the Softplus function $log(1 + exp(x))$ as a surrogate to the indicator function in computation of $l_{\theta,\delta}$ in Algorithm 1.

**Uncertainty-based Abstention Strategy** To compare **TRECA** and baseline methods under consistent abstention conditions, we conducted baseline experiments using a unified uncertainty-based abstention procedure. For baseline estimators on CATE $\hat{\tau}(x)$, following the derivation in Propo-

*Table 2.* Mean ± std of **f-regret** on retained, abstained and overall samples on real-world datasets with abstention rate = 30%.

| Method | Within-samples on Twins_mono | | | Out-of-samples on Twins_mono | | |
|---|---|---|---|---|---|---|
| | f-regret-r ↓ | f-regret-a ↓ | f-regret-o ↓ | f-regret-r ↓ | f-regret-a ↓ | f-regret-o ↓ |
| T-learner | 0.055 ± 0.121 | 0.153 ± 0.153 | 0.087 ± 0.130 | 0.061 ± 0.130 | 0.160 ± 0.152 | 0.090 ± 0.135 |
| X-learner | 0.051 ± 0.123 | 0.165 ± 0.203 | 0.078 ± 0.105 | 0.050 ± 0.130 | 0.165 ± 0.220 | 0.075 ± 0.121 |
| CFRNet | 0.018 ± 0.030 | **0.063 ± 0.065** | 0.023 ± 0.032 | 0.016 ± 0.026 | **0.065 ± 0.064** | **0.021 ± 0.027** |
| CEVAE | 0.103 ± 0.194 | 0.312 ± 0.295 | 0.177 ± 0.217 | 0.120 ± 0.206 | 0.298 ± 0.230 | 0.175 ± 0.219 |
| ESCFR | 0.091 ± 0.133 | 0.275 ± 0.166 | 0.150 ± 0.142 | 0.102 ± 0.142 | 0.283 ± 0.173 | 0.155 ± 0.156 |
| DeRCFR | 0.098 ± 0.191 | 0.470 ± 0.032 | 0.456 ± 0.114 | 0.103 ± 0.190 | 0.481 ± 0.020 | 0.460 ± 0.117 |
| DeSCN | 0.084 ± 0.166 | 0.370 ± 0.190 | 0.279 ± 0.207 | 0.087 ± 0.163 | 0.372 ± 0.196 | 0.276 ± 0.211 |
| DragonNet | 0.052 ± 0.074 | 0.189 ± 0.176 | 0.101 ± 0.127 | 0.064 ± 0.077 | 0.186 ± 0.185 | 0.103 ± 0.119 |
| CF | 0.078 ± 0.134 | 0.276 ± 0.172 | 0.197 ± 0.195 | 0.080 ± 0.142 | 0.285 ± 0.177 | 0.203 ± 0.189 |
| CFRISW | 0.025 ± 0.035 | 0.077 ± 0.050 | 0.038 ± 0.047 | 0.030 ± 0.037 | 0.084 ± 0.047 | 0.043 ± 0.184 |
| CTRL | 0.004 ± 0.001 | 0.273 ± 0.029 | 0.085 ± 0.009 | 0.006 ± 0.002 | 0.283 ± 0.040 | 0.089 ± 0.012 |
| **TRECA** | **0.001 ± 0.002** | 0.352 ± 0.171 | 0.039 ± 0.017 | **0.003 ± 0.003** | 0.322 ± 0.166 | 0.036 ± 0.017 |
| w/o mono. | 0.002 ± 0.002 | 0.333 ± 0.174 | 0.034 ± 0.020 | 0.003 ± 0.003 | 0.420 ± 0.201 | 0.045 ± 0.020 |
| w/o uncf. | 0.004 ± 0.002 | 0.106 ± 0.103 | **0.011 ± 0.020** | 0.004 ± 0.002 | 0.200 ± 0.151 | 0.023 ± 0.024 |

| Method | Within-samples on Jobs_mono | | | Out-of-samples on Jobs_mono | | |
|---|---|---|---|---|---|---|
| | f-regret-r ↓ | f-regret-a ↓ | f-regret-o ↓ | f-regret-r ↓ | f-regret-a ↓ | f-regret-o ↓ |
| T-learner | 0.195 ± 0.164 | 0.350 ± 0.371 | 0.216 ± 0.226 | 0.186 ± 0.160 | 0.361 ± 0.340 | 0.227 ± 0.193 |
| X-learner | 0.119 ± 0.162 | 0.382 ± 0.401 | 0.153 ± 0.201 | 0.112 ± 0.097 | 0.040 ± 0.051 | 0.112 ± 0.050 |
| CFRNet | 0.039 ± 0.054 | 0.082 ± 0.093 | 0.047 ± 0.061 | 0.042 ± 0.053 | **0.075 ± 0.081** | 0.047 ± 0.060 |
| CEVAE | 0.063 ± 0.070 | 0.159 ± 0.143 | 0.081 ± 0.093 | 0.074 ± 0.088 | 0.219 ± 0.160 | 0.097 ± 0.101 |
| ESCFR | 0.078 ± 0.061 | 0.173 ± 0.160 | 0.097 ± 0.088 | 0.075 ± 0.058 | 0.197 ± 0.202 | 0.103 ± 0.094 |
| DeRCFR | 0.065 ± 0.077 | 0.142 ± 0.158 | 0.087 ± 0.091 | 0.062 ± 0.083 | 0.137 ± 0.095 | 0.086 ± 0.073 |
| DESCN | 0.083 ± 0.130 | 0.204 ± 0.258 | 0.109 ± 0.188 | 0.090 ± 0.149 | 0.220 ± 0.262 | 0.116 ± 0.097 |
| DragonNet | 0.038 ± 0.058 | **0.077 ± 0.084** | 0.044 ± 0.051 | 0.042 ± 0.051 | 0.085 ± 0.088 | 0.051 ± 0.062 |
| CF | 0.112 ± 0.163 | 0.231 ± 0.270 | 0.151 ± 0.195 | 0.097 ± 0.165 | 0.235 ± 0.281 | 0.163 ± 0.204 |
| CFRISW | 0.035 ± 0.032 | 0.093 ± 0.088 | 0.045 ± 0.051 | 0.037 ± 0.042 | 0.089 ± 0.080 | 0.051 ± 0.050 |
| CTRL | 0.036 ± 0.079 | 0.097 ± 0.160 | 0.047 ± 0.092 | 0.038 ± 0.083 | 0.094 ± 0.170 | 0.053 ± 0.048 |
| **TRECA** | **0.017 ± 0.031** | 0.167 ± 0.134 | **0.033 ± 0.069** | **0.025 ± 0.038** | 0.131 ± 0.138 | **0.035 ± 0.077** |
| w/o mono. | 0.046 ± 0.053 | 0.103 ± 0.128 | 0.053 ± 0.073 | 0.043 ± 0.047 | 0.182 ± 0.137 | 0.058 ± 0.076 |
| w/o uncf. | 0.039 ± 0.048 | 0.160 ± 0.142 | 0.051 ± 0.070 | 0.043 ± 0.055 | 0.182 ± 0.142 | 0.058 ± 0.067 |

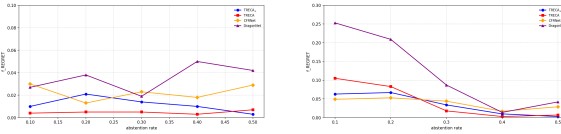

*Figure 2.* Results of f-regret on Twins_mono and Jobs_mono datasets with varying abstention rate $\delta$.

sition 3.2, we compute the conditional risk

$$\hat{V}_f(x) = \frac{1}{2} f(x)\{\theta - \hat{\rho}(x)\} + \frac{1}{2}\{\theta + (1 - 2\theta)\hat{\rho}(x)\}$$

with $\hat{\rho}(x) = \hat{\tau}(x)/2$, and abstain the samples with $(100\delta)\%$ highest conditional risk to evaluate each model solely on the remaining samples. All models were trained with the same experimental setting ($\theta = 0.5$, $\delta = 0.3$) and identical hardware configuration to ensure comparability.

### 4.2. Experimental Results

Table 2 compares the baseline models with **TRECA** and its modifications in relaxation of monotonicity (**w/o mono.**)

or unconfoundedness assumption (**w/o uncf.**), with detailed algorithms described in Algorithm 1 and 2. The results indicate that (1) our method **TRECA** stably outperforms baseline methods in **Jobs_mono** and most cases in **Twins_mono** in achieving low **f-regret-r** on retained samples, addressing the advantage of our method in learning more accurate classification rule comparing with estimation-then-decision rules derived from CATE-oriented methods (Fernández-Loría & Provost, 2022). (2) On abstained samples, **TRECA** achieves sub-optimal **f-regret-a** comparing with baseline methods. This implies a rational balance made by **TRECA** which yields accurate classification on retained samples at the cost of its performance on abstained samples. Such balance is affordable as **TRECA** generally achieves low **f-regret-o** on the entire dataset. (3) The modifications of **TRECA** achieves higher **f-regret-r** than the original version in relaxation of monotonicity and unconfoundedness while the regret is lower than those of baseline methods in general, implying the capability **TRECA** in responder classification even under the violation of these assumptions.

### 4.3. Ablation Studies

We conduct comprehensive studies to evaluate each component of our method. In the first study, we evaluate our methods in **Twins** and **Jobs** dataset when monotonicity violates. Table 3 shows that **TRECA** still induces comparatively small **f-regret-r** when monotonicity violates, while its modification **w/o mono** that does not depend on monotonicity assumption has smaller regret on datasets where monotonicity does not hold in natural. The second study aims at validating the double robustness of our methods when either model $\hat{\mu}$ or $\hat{\pi}$ is misspecified. We set $\hat{\pi}(x) \equiv 0.25$ and apply our method to classify the treatment responders. Table 4 shows that **TRECA**$^M$ trained under misspecified model still induces lower **f-regret-r** comparing with baseline method. To check the impact of abstention rate on model performances, in the third study, we shift the abstention rate $\delta$ and compare the performances of our methods with the baseline. Figure 2 manifests that under multiple choices of abstention rates ranging from 0.1 to 0.5, both **TRECA** and its modification in relaxation of monotonicity assumption denoted as **TRECA**$_+$ generally performs better in learning the classification rule on retained samples comparing with baseline methods at varying abstention rates.

*Table 3.* Results comparison (mean±std of in-sample **f-regret-r** on retained samples) among **TRECA**, its modification in relaxation of monotonicity (**w/o mono.**) and **CFRNet** on real-world datasets with abstention rate = 30%.

| Method | TRECA | w/o mono. | CFRNet |
|---|---|---|---|
| Twins | **0.006 ± 0.005** | 0.014 ± 0.022 | 0.023 ± 0.032 |
| Twins_mono | **0.001 ± 0.002** | 0.002 ± 0.002 | 0.018 ± 0.030 |
| Jobs | **0.036 ± 0.039** | 0.069 ± 0.083 | 0.081 ± 0.090 |
| Jobs_mono | **0.017 ± 0.031** | 0.046 ± 0.053 | 0.039 ± 0.054 |

*Table 4.* Results comparison (mean±std) among baseline and our method under misspecified propensity $\hat{\pi}$ (labeled by superscript $M$) on real-world datasets.

| | TRECA | TRECA$^M$ | CFRNet |
|---|---|---|---|
| Twins_mono | 0.001 ± 0.002 | 0.021 ± 0.015 | 0.023 ± 0.032 |
| Jobs_mono | 0.017 ± 0.031 | 0.035 ± 0.029 | 0.039 ± 0.054 |

### 5. Discussion

This paper proposes a theoretically-grounded framework to classify treatment responder accurately on retained samples under constraint on abstention rate. While we have discussed the adaptation of our method to possible violation of assumptions, further extension includes treatment responder classification with more complex treatments and outcomes. Another interesting problem is to consider the dual form which abstain samples as few as possible under constraint on misclassification loss, which adapts to other scenarios in which the abstention rate is at a higher priority.

## Acknowledgement

Mingming Gong was supported by ARC DP240102088 and WIS-MBZUAI 142571. Erdun Gao was supported by the Responsible AI Research Centre (RAIR). Zhiheng Zhang was supported by "the Fundamental Research Fund for the Central Universities" (Grant No,2025110602) and Independent Research Project (Grant No,2026110081) by Shanghai University of Finance and Economics. We thank the anonymous reviewers for helpful comments.

## Impact Statement

This paper presents work whose goal is to advance the field of machine learning. There are many potential societal consequences of our work, none which we feel must be specifically highlighted here.

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

# A. Additional Example and Related Work

## A.1. Additional Example on Abstention Learning

**Speed and cost balance in public policy.** The balance between speed and cost has been well addressed as an importance concern in public policy (Vlasova & Rakhmeeva, 2020). For example, during emerging pandemics, practitioners want to decide the area in which public protective policies will be implemented. In this case, treatment is whether to implement the policy in a town or community, and outcome is whether the transmission of pandemic is suppressed locally. To minimize the social impact, decision-makers want to implement public policy only if it is deterministically required. While it is much likely unnecessary to implement protective policy in remote towns with little population and obviously mandatory in metropolis in outbreak regions, there are areas where decision is highly uncertain. For this locations, abstention allows decision-maker to suspend making decision in seek of gathering more data or making more detailed assessment. Treatment responder classification in this scenario saves public welfare on costly restrictions (Vandepitte et al., 2021; Güner et al., 2021), provides a mechanism to suspend decisions for a small proportion of communities, redirecting resources toward closer evaluation on the pandemic prevalence in these areas.

## A.2. Related Work

**CATE Estimation.** CATE refers to the average treatment effect on outcomes for certain subgroup of population characterized by covariates. The most classical methods estimating CATE are based on re-weighting (Austin, 2011; Imai & Ratkovic, 2014; Fong et al., 2018). Other widely used techniques incorporate machine learning, including Causal Forests (CF) (Athey & Imbens, 2016; Wager & Athey, 2018), CEVAE (Louizos et al., 2017), etc. Methods like T-Learner and X-Learner (Künzel et al., 2019) utilize the idea of meta-learning to combine supervised learning and regression methods to enhance robustness and performance. Featured by balancing the distribution between treated and control groups, counterfactual representation learning is capable of producing precise CATE estimation with generalization guarantee, thus a popular method to estimate CATE for complex data (Johansson et al., 2016; Shalit et al., 2017; Shi et al., 2019). There have been a series of advanced CFR-based methods recently, including the combination of importance weighting (Hassanpour & Greiner, 2019), representation decomposition (Wu et al., 2022), optimal transport (Wang et al., 2023) and so on.

**Learning Individual Treatment Rules.** Treatment responder classification can be viewed as a task within learning individual treatment rules (ITR) or treatment regimes. ITR learning is featured by two types of approaches. The first approach is to perform an **estimation-then-decision** procedure which learns the decision rule based on statistical or machine-learning models on heterogeneous treatment effects, such as parametric or semi-parametric regression (Cai et al., 2010) and causal machine learning (Kreif & DiazOrdaz, 2019; Ngufor et al., 2023). The treatments are allocated through analyzing the decision curve on treatment effects in terms of individual biomarkers (Vickers & Elkin, 2006), and the uncertainty of the learned policy can be drawn by constructing simultaneous confidence band on the decision curve (Zhou & Ma, 2012; Ma & Zhou, 2017; Guo et al., 2021). The second approach is to treat the problem as a **policy learning** task, directly optimizing the decision rule through regret minimization or reward maximization, which is typically equivalent to a weighted classification problem (Zhao et al., 2012). Some methods separated the treatment effect from the reward, and considered linear, kernel-based (Huang & Fong, 2014) or tree based (Zhu et al., 2018) methods to discover the decision boundary of the treatment rule. Moreover, taking the robustness into account, (Pan & Zhao, 2021) proposed a doubly robust estimator to learn the individual treatment rule, and (Mo et al., 2021) proposed a distributionally robust method to deal with the situation with training and testing data are not identical. Our work lies in this type of approach by dealing with the binary misclassification loss in selecting treatment responders. Under this scenario, the conditional misclassification risk involves joint distribution of potential outcomes, inducing extra justification on identification and construction of doubly robust estimators.

**Abstention Learning.** Abstention learning allows models to abstain from making decisions on challenging instances to avoid costly errors. The roots of abstention learning can be traced back to (Chow, 1957; 1970), which introduced the fundamental trade-off between misclassification error and rejection rate in binary classification. Abstention learning framework generally consists of a predictor which predicts the outcome label, and a rejector choosing which samples to abstain. Cortes et al. (2016); Mao et al. (2024) focus on cost-based abstention, which minimizes the loss function balanced between cost and accuracy. A challenge in cost-based abstention learning is that there is a lack of criterion to determine the cost of abstention. Moreover, since the optimal rule generally involves confidence-cost comparison (Cortes et al., 2024), it is difficult to scale cost and confidence in the same magnitude. Other works focus on the confidence-based or interval-based

rejection rule, which abstains when the minimum deviation on misclassification probability exceeds a predefined confidence threshold or interval (Herbei & Wegkamp, 2006; Denis & Hebiri, 2020; Zhu & Nowak, 2022). However, these methods do not explicitly control the rejection rate, i.e., proportion of samples being abstained, which is a key feature to interpret the abstention strategy. Incorporating treatment responder classification with other types of abstention therefore helps increase the interpretability and broaden the application scenarios of the framework.

**Deferral System.** Deferral system or deferral policy learning deals with a similar scenario as abstention learning, in which uncertain decisions are delegated to a secondary agent which is typically a human (Gao & Yin, 2025). Palomba et al. (2025); Ghoummaid & Shalit (2024) apply causal framework on the deferral system, with a similar goal of minimizing risk under constraint on the proportion of deferred / abstained samples. However, a key difference is that our work focuses on treatment responder selection which involves further discussion on identification and development on robust methods, while Gao & Yin (2025); Palomba et al. (2025) aim at learning the causal effect under RD design or outcome reward maximization.

## B. Key Notations

*Table 5.* Summary of Notations.

| Notation | Meaning |
|---|---|
| $\min, \max$ | Minimum / Maximum operator. |
| $[n]$ | Index set $\{1, \ldots, n\}$. |
| $O_p(\cdot)$ | Probabilistic order. |
| $o_p(\cdot)$ | Converge in probability. |
| $\mathcal{F}$ | Hypothesis class for the predictor $f$. |
| $\mathcal{R}$ | Hypothesis class for the rejector $r$. |
| $f(x)$ | Predictor determining whether an individual is a treatment responder. |
| $r(x)$ | Rejector (abstention rule); $r(x) = 1$ indicates abstention. |
| $T$ | Binary treatment assignment. |
| $Y$ | Observed outcome. |
| $Y(1), Y(-1)$ | Potential outcomes under treatment and control. |
| $R$ | Responder indicator. |
| $\tau(x)$ | Conditional Average Treatment Effect (CATE). |
| $\rho(x)$ | Conditional responding probability. |
| $\pi(x)$ | Propensity score. |
| $V_f(x)$ | Conditional risk. |
| $\beta^*$ | Optimal abstention threshold (CVaR-based). |
| $(u)^- = \min(u, 0)$ | Negative part operator. |
| $I(\cdot)$ | Indicator function. |
| $F_{V_f}$ | Cumulative distribution function of conditional risk. |
| $\hat{\tau}, \hat{\mu}, \hat{\pi}$ | Estimated CATE, outcome regression, and propensity score. |
| $\delta$ | Abstention rate. |
| $\theta$ | Weighting parameter on misclassification types. |

## C. Algorithms on TRECA and its modification

The algorithm of **TRECA** is provided in Algorithm 1. Its modification in relaxation of monotonicity assumption is provided in Algorithm 2

## D. Supplementary Proofs and Theorems

### D.1. Proof on Proposition 2.6

Proposition 2.6 is a direct result from Theorem 2 in Rockafellar et al. (2000).

---

**Algorithm 1** **T**reatment **RE**sponder **C**lassification with **A**bstention under Monotonicity Assumption (**TRECA**)

---

**Input:** Level $\alpha \in (0, 1)$, $k, m \in \mathbb{N}$, data $\mathcal{D} = \{(X_i, T_i, Y_i) : i = 1, \ldots, n\}$, initialized $\hat{\mu}, \hat{\pi}, \hat{\tau}$.
Estimate $\hat{\pi}, \hat{\mu}$ through minimizing MSE $\frac{1}{n} \sum_{i=1}^{n} (T_i - \hat{\pi}(X_i))^2$ and $\frac{1}{n} \sum_{i=1}^{n} (Y_i - \hat{\mu}_{T_i}(X_i))^2$. Let $\omega_f$ and $\omega_\tau$ be the parameters of $\hat{f}$ and $\hat{\tau}$.
**while** not converge **do**
    Sample a mini-batch of size $k$ with index $K = \{i_1, \ldots, i_k\}$ from $\mathcal{D}$.
    $s = 0$.
    **for** $s \leq m$ **do**
        Update $\hat{\beta} = \hat{\beta} + \epsilon_\beta \frac{1}{k} \sum_{i \in K} \partial_\beta l_{\theta,\delta}(X_i, T_i, Y_i, \hat{f}; \hat{\tau}, \hat{\mu}, \hat{\pi}, \beta)$.
        Update $\omega_\tau = \omega_\tau - \epsilon_\tau \frac{1}{k} \sum_{i \in K} \partial_{\omega_\tau} l_{\theta,\delta}(X_i, T_i, Y_i, \hat{f}; \hat{\tau}, \hat{\mu}, \hat{\pi}, \hat{\beta})$.
        Update $\omega_f = \omega_f - \epsilon_f \frac{1}{k} \sum_{i \in K} \partial_{\omega_f} l_{\theta,\delta}(X_i, T_i, Y_i, \hat{f}; \hat{\tau}, \hat{\mu}, \hat{\pi}, \hat{\beta})$.
        $s = s + 1$.
    **end for**
**end while**
Compute $V_{\hat{f}}(X_i)$ from Proposition 2.6.
Set $\hat{q}_\delta = \inf\{q : \frac{1}{n} \sum_{i=1}^{n} I(V_{\hat{f}}(X_i) \geq q) - \delta \leq 0\}$.
Estimate the abstention rule $\hat{r}(X_i) = I(V_{\hat{f}}(X_i, \hat{\tau}) > \hat{q}_\delta)$.

---

### D.2. Proof on Proposition 3.2

Recall that $\rho(X) = \mathbb{P}(R = 1 \mid X = x)$, the proposition is an immediate result from

$$
\begin{aligned}
V_f(x) =& \theta \cdot \mathbb{E}[I(f(X) = +1)I(R = -1) \mid X = x] + (1 - \theta) \cdot \mathbb{E}[I(f(X) = -1)I(R = +1) \mid X = x] \\
=& \frac{\theta}{2}(1 + f(x))(1 - \rho(x)) + \frac{1 - \theta}{2}(1 - f(x))\rho(x) \\
=& \frac{1}{2} f(x)(\theta - \rho(x)) + \frac{1}{2}(\theta + (1 - 2\theta)\rho(x))
\end{aligned}
$$

and

$$
\begin{aligned}
\tau(x) =& \mathbb{E}(Y(1) - Y(-1) \mid X = x) \\
=& 2\mathbb{P}(Y(-1) = -1, Y(+1) = +1 | X = x) \\
& - 2\mathbb{P}(Y(-1) = +1, Y(+1) = -1 | X = x) \\
=& 2\rho(X = x) - 2\mathbb{P}(Y(-1) < Y(+1) | X = x).
\end{aligned}
$$

### D.3. Proof on Proposition 3.4

From Proposition 3.2, we can rewrite

$$
\begin{aligned}
V_f(x) =& \frac{1}{2} f(x)(\theta - \rho(x)) + \frac{1}{2}(\theta + (1 - 2\theta)\rho(x)) \\
=& \frac{1 - 2\theta - f(x)}{2}\rho(x) + \frac{\theta}{2}(f(x) + 1).
\end{aligned}
$$

Since $f(x) \in \{-1, 1\}$ and $\theta \in [0, 1]$, $1 - 2\theta - f(x) \geq 0$ if $f(x) = -1$, and $1 - 2\theta - f(x) \leq 0$ if $f(x) = 1$. Therefore,

$$
V_f(x) \leq \begin{cases} \frac{1}{2}(-1) \cdot (\theta - \rho^U(x)) + \frac{1}{2}(\theta + (1 - 2\theta)\rho^U(x)) = (1 - \theta)\rho^U(x), & f(x) = -1; \\ \frac{1}{2}(\theta - \rho^L(x)) + \frac{1}{2}(\theta + (1 - 2\theta)\rho^L(x)) = \theta(1 - \rho^L(x)), & f(x) = 1, \end{cases}
$$

which proofs Proposition 3.4.

---

**Algorithm 2** **T**reatment **RE**sponder **C**lassification with **A**bstention without Monotonicity Assumption (**w/o mono.**)

---

**Input:** Level $\alpha \in (0, 1)$, $k, m \in \mathbb{N}$, data $\mathcal{D} = \{(X_i, T_i, Y_i) : i = 1, \ldots, n\}$, initialized $\hat{\mu}, \hat{\pi}, \hat{\tau}$.

Estimate $\hat{\pi}, \hat{\mu}$ through minimizing MSE $\frac{1}{n} \sum_{i=1}^{n} (T_i - \hat{\pi}(X_i))^2$ and $\frac{1}{n} \sum_{i=1}^{n} (Y_i - \hat{\mu}_{T_i}(X_i))^2$. Let $\omega_f$ and $\omega_\tau$ be the parameters of $\hat{f}$ and $\hat{\tau}$.

**while** not converge **do**

    Sample a mini-batch of size $k$ with index $K = \{i_1, \ldots, i_k\}$ from $\mathcal{D}$.

    $s = 0$.

    **for** $s \leq m$ **do**

        Update $\hat{\beta} = \hat{\beta} + \epsilon_\beta \frac{1}{k} \sum_{i \in K} \partial_\beta m_{\theta, \delta}(X_i, T_i, Y_i, \hat{f}; \hat{\tau}, \hat{\mu}, \hat{\pi}, \beta)$.

        Update $\omega_\tau = \omega_\tau - \epsilon_\tau \frac{1}{k} \sum_{i \in K} \partial_{\omega_\tau} m_{\theta, \delta}(X_i, T_i, Y_i, \hat{f}; \hat{\tau}, \hat{\mu}, \hat{\pi}, \hat{\beta})$.

        Update $\omega_f = \omega_f - \epsilon_f \frac{1}{k} \sum_{i \in K} \partial_{\omega_f} m_{\theta, \delta}(X_i, T_i, Y_i, \hat{f}; \hat{\tau}, \hat{\mu}, \hat{\pi}, \hat{\beta})$.

        $s = s + 1$.

    **end for**

**end while**

Compute $U_{\hat{f}}(X_i)$ from Proposition 2.6.

Set $\hat{q}_\delta = \inf\{q : \frac{1}{n} \sum_{i=1}^{n} I(U_{\hat{f}}(X_i) \geq q) - \delta \leq 0\}$.

Estimate the abstention rule $\hat{r}(X_i) = I(U_{\hat{f}}(X_i, \hat{\tau}) > \hat{q}_\delta)$.

---

### D.4. Proof on Theorem 2.5

Note that

$$
\begin{aligned}
&\tilde{L}_\theta(f, r) \\
&= \theta \mathbb{P}(f(X) = 1, R = -1, r(X) = 0) + (1 - \theta) \mathbb{P}(f(X) = -1, R = 1, r(X) = 0) \\
&= \mathbb{E}\left[\theta I(f(X) = 1, R = -1) + (1 - \theta) I(f(X) = -1, R = 1) \mid r(X) = 0\right] \mathbb{P}(r(X) = 0) \\
&= \int_{x \in \mathcal{X}} \mathbb{E}\left[\theta I(f(X) = 1, R = -1) + (1 - \theta) I(f(X) = -1, R = 1) \mid X = x\right] \times \\
&\quad p(x \mid r(X) = 0) \mathbb{P}(r(X) = 0) dx \\
&= \int_{x \in \mathcal{X}} V_f(x) p(x \mid r(X) = 0) \mathbb{P}(r(X) = 0) dx,
\end{aligned}
$$

which is the integral on conditional probability density function $p(x \mid r(X) = 0)$ within retained samples weighted by $V_f(x)$. It is straightforward that under constraint $E[r(X)] \leq \delta$, such weighted integral is minimized at $r^*(X) = I(V_f(X) > F_{V_f}^{-1}(1 - \delta))$. For $r = r^*$, we have

$$
\begin{aligned}
&\tilde{L}_\theta(f, r) = L_{\theta, \delta}(f) \\
&= \int_{x \in \mathcal{X}} \mathbb{E}\left\{\theta I(f(X) = 1, R = -1) + (1 - \theta) I(f(X) = -1, R = 1) \mid X = x, V_f(x) \leq F_{V_f}^{-1}(1 - \alpha)\right\} \times \\
&\quad p(x \mid V_f(x) \leq F_{V_f}^{-1}(1 - \alpha)) \, dx \\
&= \int_{x \in \mathcal{X}} \mathbb{E}\left\{\theta I(f(X) = 1, R = -1) + (1 - \theta) I(f(X) = -1, R = 1) \mid X = x\right\} \times \\
&\quad p(x \mid V_f(x) \leq F_{V_f}^{-1}(1 - \alpha)) \, dx \\
&= \int_{x \in \mathcal{X}} V_f(x) \cdot p(x \mid V_f(x) \leq F_{V_f}^{-1}(1 - \alpha)) \, dx \\
&= \mathbb{E}\left\{V_f(x) \mid V_f(x) \leq F_{V_f}^{-1}(1 - \alpha)\right\}.
\end{aligned}
$$

where $p(x \mid \cdot)$ is the conditional probability density function.

## D.5. Proof on Theorem 3.3

The Proof on Theorem 3.3 is a modification from the proofs in Kallus (2023) by adopting different target functions and scrutinizing the differences in objects that CVaR is taken upon. In the following proof, denote $\hat{\mathbb{E}}$ as the sample average. Recall that $(\tau, \mu, \pi)$ are the true values, and $\beta^* = \arg\max_{\beta \in \mathbb{R}} \mathbb{E} l_{\theta,\delta}(f; \tau, \mu, \pi, \beta)$. Note that the following expansion holds

$$\hat{L}_{\theta,\delta}(f; \hat{\tau}, \hat{\mu}, \hat{\pi}) - L_{\theta,\delta}(f)$$
$$= \hat{\mathbb{E}} l_{\theta,\delta}(X, T, Y, f; \hat{\tau}, \hat{\mu}, \hat{\pi}, \hat{\beta}) - \mathbb{E} l_{\theta,\delta}(X, T, Y, f; \tau, \mu, \pi, \beta^*)$$
$$= \mathbb{E}(l_{\theta,\delta}(X, T, Y, f; \hat{\tau}, \hat{\mu}, \hat{\pi}, \hat{\beta}) - l_{\theta,\delta}(X, T, Y, f; \tau, \mu, \pi, \beta^*)) \tag{8}$$
$$+ (\hat{\mathbb{E}} - \mathbb{E})(l_{\theta,\delta}(X, T, Y, f; \hat{\tau}, \hat{\mu}, \hat{\pi}, \hat{\beta}) - l_{\theta,\delta}(X, T, Y, f; \tau, \mu, \pi, \hat{\beta})) \tag{9}$$
$$+ (\hat{\mathbb{E}} - \mathbb{E})(l_{\theta,\delta}(X, T, Y, f; \tau, \mu, \pi, \hat{\beta}) - l_{\theta,\delta}(X, T, Y, f; \tau, \mu, \pi, \beta^*)), \tag{10}$$

It suffices to bound each term separately. To start with, observe that

$$\mathbb{E} l_{\theta,\delta}(X, T, Y, f; \hat{\tau}, \hat{\mu}, \hat{\pi}, \hat{\beta})$$
$$= \hat{\beta} + \frac{1}{1-\delta} \mathbb{E} I(V_f(X; \hat{\tau}) \le \hat{\beta}) \Big[ \frac{1}{2}(f(X) + 1)\theta + \Big\{ \frac{1}{4} - \frac{\theta}{2} - \frac{1}{4}f(X) \Big\} \times$$
$$\Big\{ \hat{\mu}_1(X) - \hat{\mu}_{-1}(X) + \frac{\pi(X)}{\hat{\pi}(X)}(\mu_1(X) - \hat{\mu}_1(X)) - \frac{1 - \pi(X)}{1 - \hat{\pi}(X)}(\mu_{-1}(X) - \hat{\mu}_{-1}(X)) \Big\} - \hat{\beta} \Big]. \tag{11}$$

We write $A_f(X) = \frac{1}{2}(f(X) + 1)\theta$ and $B_f(X) = \frac{1}{4} - \frac{\theta}{2} - \frac{1}{4}f(X)$ in short. For (8), note that

$$|\mathbb{E}(l_{\theta,\delta}(X, T, Y, f; \hat{\tau}, \hat{\mu}, \hat{\pi}, \hat{\beta}) - l_{\theta,\delta}(X, T, Y, f; \tau, \mu, \pi, \hat{\beta}))|$$
$$\le |\mathbb{E}(l_{\theta,\delta}(X, T, Y, f; \hat{\tau}, \hat{\mu}, \hat{\pi}, \hat{\beta}) - l_{\theta,\delta}(X, T, Y, f; \hat{\tau}, \hat{\mu}, \pi, \hat{\beta}))|$$
$$+ |\mathbb{E}(l_{\theta,\delta}(X, T, Y, f; \hat{\tau}, \hat{\mu}, \pi, \hat{\beta}) - l_{\theta,\delta}(X, T, Y, f; \hat{\tau}, \mu, \pi, \hat{\beta}))| \tag{12}$$
$$+ |\mathbb{E}(l_{\theta,\delta}(X, T, Y, f; \hat{\tau}, \mu, \pi, \hat{\beta}) - l_{\theta,\delta}(X, T, Y, f; \tau, \mu, \pi, \hat{\beta}))|$$
$$+ |\mathbb{E}(l_{\theta,\delta}(X, T, Y, f; \tau, \mu, \pi, \hat{\beta}) - l_{\theta,\delta}(X, T, Y, f; \tau, \mu, \pi, \beta))|.$$

We now bound each term in (12) sequentially. For the first term, note that $|B_f(X)| \le \frac{1-\theta}{2}$,

$$|\mathbb{E}(l_{\theta,\delta}(X, T, Y, f; \hat{\tau}, \hat{\mu}, \hat{\pi}, \hat{\beta}) - l_{\theta,\delta}(X, T, Y, f; \hat{\tau}, \hat{\mu}, \pi, \hat{\beta}))|$$
$$\le \frac{1}{1-\delta} \mathbb{E} \Big[ I[V_f(X; \hat{\tau}) \le \hat{\beta}] \frac{B_f(X)}{\hat{\pi}(X)} |\pi(X) - \hat{\pi}(X)||\mu_1(X) - \hat{\mu}_1(X)| \Big]$$
$$+ \frac{1}{1-\delta} \mathbb{E} \Big[ I[V_f(X; \tau) \le \hat{\beta}] \frac{B_f(X)}{1 - \hat{\pi}(X)} |\pi(X) - \hat{\pi}(X)||\mu_0(X) - \hat{\mu}_0(X)| \Big]$$
$$\le \frac{1 - \theta}{2(1-\delta)\underline{\pi}} \|\pi - \hat{\pi}\|_2 (\|\mu_1 - \hat{\mu}_1\|_2 + \|\mu_0 - \hat{\mu}_0\|_2).$$

Meanwhile, note that the second term equals zero. For the third term, set $t = \|V_f(X; \tau) - V_f(X; \hat{\tau})\|_q^{\frac{q}{1+q}}$. Note that

$$\|V_f(X; \tau) - V_f(X; \hat{\tau})\|_q \le \frac{1 - \theta}{2} \|\tau - \hat{\tau}\|_q,$$

hence $t \le O_p(\epsilon_n)$ from the convergence of CATE estimator $\|\hat{\tau} - \tau\|_q = O_p(\epsilon_n^{\frac{q}{q+1}})$ for some $q > 1$. From Assumption 2.4, there exists $\epsilon$ and $c > 0$ such that $V_f(X; \tau)$ has density bounded by $c$ in the local area of $[\beta^* - \epsilon, \beta^* + \epsilon]$. From Lemma 1 and Lemma EC.2 in Kallus (2023),

$$|\hat{\beta} - \beta^*| = O_p(\|V_f(X; \tau) - V_f(X; \hat{\tau})\|_q^{\frac{q}{q+1}} + n^{-1/2}) = O_p(\|\hat{\tau} - \tau\|_q^{\frac{q}{q+1}} + n^{-1/2}) = O_p(\epsilon_n^{1/2} \vee n^{-1/2}).$$

Therefore, $(\hat{\beta} - \beta^*)^2 = O_p(\epsilon_n \vee n^{-1})$, and $V_f(X; \tau) - \hat{\beta}$ has density bounded by $c$ in $[-t, t]$ for sufficiently large $n$. Applying Hölder's inequality we have

$$
\begin{aligned}
&|\mathbb{E}(l_{\theta,\delta}(X, T, Y, f; \hat{\tau}, \mu, \pi, \hat{\beta}) - l_{\theta,\delta}(X, T, Y, f; \tau, \mu, \pi, \hat{\beta}))| \\
=&\frac{1}{1-\delta}\left|\mathbb{E}\left[(I[V_f(X; \hat{\tau}) \leq \hat{\beta}] - I[V_f(X; \tau) \leq \hat{\beta}])(V_f(X; \tau) - \hat{\beta})\right]\right| \\
\leq&\frac{1}{1-\delta}\mathbb{E}\left[|I[V_f(X; \hat{\tau}) \leq \hat{\beta}] \neq I[V_f(X; \tau) \leq \hat{\beta}]||V_f(X; \tau) - \hat{\beta}|\right] \\
\leq&\frac{1}{1-\delta}\mathbb{E}[|V_f(X; \tau) - V_f(X; \hat{\tau})|I[|V_f(X; \tau) - \hat{\beta}| \leq t]] \\
&+ \frac{1}{1-\delta}\mathbb{E}[|V_f(X; \tau) - V_f(X; \hat{\tau})|I[|V_f(X; \tau) - V_f(X; \hat{\tau})| > t]] \\
\leq&\frac{1}{1-\delta}\|V_f(X; \tau) - V_f(X; \hat{\tau})\|_q \mathbb{P}(|V_f(X; \tau) - \hat{\beta}| \leq t)^{(q-1)/q} \\
&+ \frac{1}{1-\delta}\|V_f(X; \tau) - V_f(X; \hat{\tau})\|_q \mathbb{P}(|V_f(X; \tau) - V_f(X; \hat{\tau})| > t)^{(q-1)/q} \\
\leq&\frac{1}{1-\delta}\|V_f(X; \tau) - V_f(X; \hat{\tau})\|_q (2ct)^{(q-1)/q} + \frac{1}{1-\delta}\|V_f(X; \tau) - V_f(X; \hat{\tau})\|_q^q t^{1-q} \\
=&\frac{1}{1-\delta}((2c)^{(q-1)/q} + 1)\|V_f(X; \tau) - V_f(X; \hat{\tau})\|_q^{\frac{2q}{1+q}} \\
\leq&\frac{1}{1-\delta}((2c)^{(q-1)/q} + 1)(\frac{1-\theta}{2})^{\frac{2q}{1+q}}\|\tau - \hat{\tau}\|_q^{\frac{2q}{q+1}}.
\end{aligned}
$$

For the fourth term, let $g(\beta) = \mathbb{E}l_{\theta,\delta}(X, T, Y, f; \tau, \mu, \pi, \beta)$. From (11), $g'(\beta^*) = 0$ and $|g''(\beta)| \leq \frac{1}{1-\delta}(F'_{V_f}(F_{V_f}^{-1}(1 - \delta)) + 1) \leq \frac{1+c}{1-\delta}$ on $\beta \in [\beta^* - \epsilon, \beta^* + \epsilon]$. Therefore, from Taylor's expansion we have

$$
|\mathbb{E}(l_{\theta,\delta}(X, T, Y, f; \tau, \mu, \pi, \hat{\beta}) - l_{\theta,\delta}(X, T, Y, f; \tau, \mu, \pi, \beta^*))| \leq \frac{1+c}{2(1-\delta)}(\hat{\beta} - \beta^*)^2.
$$

From the deduction above which bounds each term in (8) respectively, there exists constants $c_1, c_2, c_3 > 0$ such that

$$
\begin{aligned}
&|\mathbb{E}(l_{\theta,\delta}(X, T, Y, f; \hat{\tau}, \hat{\mu}, \hat{\pi}, \hat{\beta}) - l_{\theta,\delta}(X, T, Y, f; \tau, \mu, \pi, \hat{\beta}))| \\
\leq&c_1\|\pi - \hat{\pi}\|_2\|\mu - \hat{\mu}\|_2 + c_2\|\tau - \hat{\tau}\|_q^{\frac{2q}{q+1}} + c_3(\hat{\beta} - \beta^*)^2.
\end{aligned}
\tag{13}
$$

Combining the convergence rates on nuisance parameters in Theorem 3.3, we have the following bound on term (8)

$$
\mathbb{E}(l_{\theta,\delta}(X, T, Y, f; \hat{\tau}, \hat{\mu}, \hat{\pi}, \hat{\beta}) - l_{\theta,\delta}(X, T, Y, f; \tau, \mu, \pi, \beta^*)) = o_p(\epsilon_n \vee n^{-1/2}).
$$

Following the same technic bounding (EC.15) and (EC.16) in Kallus (2023), we have (9), (10) are both $o_p(n^{-1/2})$. Combining Equation (8)-(10) we have $\hat{L}_{\theta,\delta}(f; \hat{\tau}, \hat{\mu}, \hat{\pi}) - L_{\theta,\delta}(f) = O_p(\epsilon_n \vee n^{-1/2})$.

### D.6. Proof on Theorem 3.6

Let $\hat{\beta}_U = \arg\max_{\beta \in \mathbb{R}} \mathbb{E}m_{\theta,\delta}(f; \hat{\tau}, \hat{\mu}, \hat{\pi}, \beta)$ and note that $\beta^* = \arg\max_{\beta \in \mathbb{R}} \mathbb{E}I[V_f(X) \leq \beta](V_f(X) - \beta)$. From Assumption 2.4 there exists $c > 0$ such that the probability density function of $V_f(X)$ is bounded by $c$. Let $c_\rho$ be the upper bound on probability density function of $\rho(X)$ stated in condition (a). Take $t = \|V_f(X) - U_f(X; \hat{\tau})\|_q^{q/(q+1)}$ and observe that

$$
\|V_f(X) - U_f(X; \hat{\tau})\|_q \leq \|\frac{1 - f(X) - 2\theta}{2} \cdot \max\{\rho(X) - \hat{\rho}^L(X), \hat{\rho}^U(X) - \rho(X)\}\|_q \leq (1 - \theta)d_\infty.
$$

Hence for any fixed $\beta \in \mathbb{R}$ and $f \in \mathcal{F}$, we have

$$
\begin{aligned}
&\mathbb{P}(I[V_f(X) \leq \beta] \neq I[U_f(X;\hat{\tau}) \leq \beta]) \\
&\leq \mathbb{P}(|V_f(X) - \beta| \leq t) + \mathbb{P}(I[U_f(X;\hat{\tau}) \leq \beta] \neq I[V_f(X) \leq \beta], |V_f(X) - \beta| > t) \\
&\leq 2tc + \mathbb{P}(|V_f(X) - U_f(X;\hat{\tau})| > t) \\
&\leq 2tc + \|V_f(X) - U_f(X;\hat{\tau})\|_q^q t^{-q} \\
&= (2c + 1)\|V_f(X) - U_f(X;\hat{\tau})\|_q^{\frac{q}{q+1}}. \\
&\leq (2c + 1)\|V_f(X) - U_f(X;\hat{\tau})\|_q^{\frac{2q}{q+1}} \leq (2c + 1)((1-\theta)d_\infty)^{\frac{2q}{q+1}}.
\end{aligned}
\tag{14}
$$

Moreover, from Lemma EC.2 in Kallus (2023),

$$
|\hat{\beta}_U - \beta^*| \leq O_p(\|V_f(X) - U_f(X;\hat{\tau})\|_q^{\frac{q}{q+1}} + n^{-1/2}) = O_p((d_\infty)^{\frac{q}{q+1}} + n^{-1/2}).
$$

Therefore,

$$
\begin{aligned}
&\mathbb{P}(I[V_f(X) \leq \beta^*] \neq I[V_f(X) \leq \hat{\beta}_U]) \\
&\leq \mathbb{P}(|V_f(X) - \beta^*| \leq |\hat{\beta}_U - \beta^*|) \\
&\leq 2c|\hat{\beta}_U - \beta^*| = O((d_\infty)^{\frac{q}{q+1}} + n^{-1/2}).
\end{aligned}
\tag{15}
$$

Combining (14) with $\beta = \hat{\beta}_U$ and (15), we have

$$
\begin{aligned}
&\mathbb{P}(I[V_f(X) \leq \beta^*] \neq I[U_f(X;\hat{\tau}) \leq \hat{\beta}_U]) \\
&\leq \mathbb{P}(I[V_f(X) \leq \hat{\beta}_U] \neq I[U_f(X;\hat{\tau}) \leq \hat{\beta}_U]) + \mathbb{P}(I[V_f(X) \leq \beta^*] \neq I[V_f(X) \leq \hat{\beta}_U]) \\
&= O((d_\infty)^{\frac{q}{q+1}} \vee n^{-1/2}).
\end{aligned}
\tag{16}
$$

From the expression of $V_f(X)$ in Proposition 3.2, $f^*(x) = 2I[\rho(x) > \theta] - 1$,[2] and from condition *(b)* $\hat{f}_M(x) = -1$ if $\rho^U(x;\hat{\tau}) \leq \theta$, $\hat{f}_M(x) = 1$ if $\rho^L(x;\hat{\tau}) > \theta$. Consequently,

$$
\begin{aligned}
&\mathbb{P}(f^*(X) \neq \hat{f}_M(X)) \\
&\leq \mathbb{P}(\rho(x) \leq \theta, \hat{\rho}^L(x) > \theta) + \mathbb{P}(\rho(x) > \theta, \hat{\rho}^U(x) \leq \theta) + \mathbb{P}(\hat{\rho}^L(x) < \theta < \hat{\rho}^U(x)) \\
&\leq \mathbb{P}(|\rho(x) - \theta| < d_\infty) + d_\infty \leq (c_\rho + 1)d_\infty,
\end{aligned}
$$

where $c_\rho$ is the upper bound on the density of $\rho(X)$. Hence

$$
\mathbb{P}(I[V_{f^*}(X) \leq \beta^*] \neq I[V_{\hat{f}_M}(X) \leq \beta^*]) \leq \mathbb{P}(f^*(X) \neq \hat{f}_M(X)) \leq (c_\rho + 1)d_\infty.
\tag{17}
$$

Therefore combining (16) and (17), we finally get

$$
\begin{aligned}
&\mathbb{P}(r^*(X) \neq \hat{r}_M(X)) \\
&= \mathbb{P}(I[V_{f^*}(X) \leq \beta^*] \neq I[U_{\hat{f}_M}(X;\hat{\tau}) \leq \hat{\beta}_U]) \\
&\leq \mathbb{P}(I[V_{f^*}(X) \leq \beta^*] \neq I[V_{\hat{f}_M}(X) \leq \beta^*]) \\
&\quad + \mathbb{P}(I[V_{\hat{f}_M}(X) \leq \hat{\beta}_U] \neq I[U_{\hat{f}_M}(X;\hat{\tau}) \leq \hat{\beta}_U]) + \mathbb{P}(I[V_{\hat{f}_M}(X) \leq \beta^*] \neq I[V_{\hat{f}_M}(X) \leq \hat{\beta}_U]) \\
&\leq O(d_\infty) + O((d_\infty)^{\frac{q}{q+1}} \vee n^{-1/2}) = O(d_\infty \vee n^{-1/2}),
\end{aligned}
$$

which completes the proof.

---

[2]To be rigorous, $f^* \in \mathcal{F}^{opt}$, where $\mathcal{F}^{opt}$ is the set of predictors at minimization of $L_{\theta,\delta}(f)$.

**D.7. Proof on Theorem 3.5**

We proceed similar as in Kallus (2023)by checking the orthogonality of parameters $(\tau, \mu, \pi, \beta)$ ensuing double robustness. Note that

$$\mathbb{E}m_{\theta,\delta}(X, T, Y, f; \hat{\tau}, \hat{\mu}, \hat{\pi}, \hat{\beta})$$
$$=\hat{\beta} + \frac{1}{1-\delta}\mathbb{E}I(U_f(X; \hat{\tau}) \le \hat{\beta})\Big[U_f(X; \hat{\tau}) - \hat{\beta}$$
$$+ \partial_\tau U_f(X; \hat{\tau})\left\{\frac{\pi(X)}{\hat{\pi}(X)}(\mu_1(X) - \hat{\mu}_1(X)) - \frac{1 - \pi(X)}{1 - \hat{\pi}(X)}(\mu_{-1}(X) - \hat{\mu}_{-1}(X))\right\}\Big].$$

The double robustness in $e, \mu$ comes from the observation that $\mathbb{E}m_{\theta,\delta}(X, T, Y, f; \hat{\tau}, \mu, \pi, \hat{\beta})$ is a combination of CATE weighted by fixed weight function in $X$ given $\hat{\tau}, \hat{\beta}$. The orthogonality of $\tau, \beta^*$ comes from the fact that let $h(\hat{\tau}, \hat{\beta}) = \mathbb{E}m_{\theta,\delta}(X, T, Y, f; \hat{\tau}, \mu, \pi, \hat{\beta}) = \hat{\beta} + \frac{1}{1-\delta}\mathbb{E}I(U_f(X; \hat{\tau}) \le \hat{\beta})(U_f(X; \hat{\tau}) - \hat{\beta})$. Then from the same technic bounding the third the fourth term in (12), there exists constants $c_1', c_2' > 0$ such that

$$|h(\hat{\tau}, \hat{\beta}) - h(\tau, \beta^*)| \le c_1'\|\hat{\tau} - \tau\|_q^{\frac{2q}{q+1}} + c_2'(\hat{\beta} - \beta^*)^2.$$

Therefore, $h(\hat{\tau}, \hat{\beta})$ has zero derivative at $(\tau, \beta^*)$ and the orthogonality holds following the same argument after (14) in Kallus (2023). In conclusion, the double robustness and ensuing orthogonality continues to hold for upper bound estimator (6).

**D.8. Generalization Bound on Misclassification Loss with Abstention under Monotonicity**

We derive the generalization bound on the naive estimator $\hat{L}_{\theta,\delta}^{\text{naive}}(f) = \sup_{\beta \in \mathbb{R}}\{\beta + \frac{1}{n(1-\delta)}\sum_{i=1}^n(V_f(X_i) - \beta)_-\}$ on loss function $L_{\theta,\delta}(f)$ measuring the complexity function class using Empirical Rademacher Complexity in Definition D.1.

**Definition D.1** (Empirical Rademacher Complexity). Let $\mathcal{F}$ be a family of functions mapping from $\mathcal{X}$ to $\{-1, +1\}$. Then, for a fixed set $S = (x_1, \ldots, x_n)$ of size $n$ sampled from $\mathcal{X}$, the empirical Rademacher complexity of $\mathcal{F}$ with respect to the sample $S$ is

$$\mathfrak{R}_n(\mathcal{F}) = \mathbb{E}_\sigma\left[\sup_{f \in \mathcal{F}} \frac{1}{n}\left|\sum_{i=1}^n \sigma_i f(x_i)\right|\right],$$

where $\sigma = (\sigma_1, \ldots, \sigma_n)^\top$, with $\sigma_i$'s be independent identically distributed random variables taking values in $\{-1, +1\}$ with fair probability.

**Theorem D.2** (Generalization Bound on Surrogate Loss). *Let $\mathfrak{R}_n(\mathcal{H})$ be the empirical Rademacher complexity for a function class $\mathcal{H}$ with respect to the training samples $\mathcal{D}$ with size $n$. Then under Assumption 3.1, there exists $c = \max_{x \in \mathcal{X}} \frac{1}{2\delta}|\theta - \frac{\tau(x)}{2}|$ such that with probability at least $1 - \epsilon$,*

$$L_{\theta,\delta}(f) \le \min_{f \in \mathcal{F}} \hat{L}_{\theta,\delta}^{naive}(f) + 2c\mathfrak{R}_n(\mathcal{F}) + 5c\sqrt{\frac{2\log\frac{8}{\epsilon}}{n}}.$$

PROOF ON THEOREM D.2

Let $\beta^* = \arg\max_{\beta \in \mathbb{R}} g_{\theta,\delta}(f; \beta)$. with $g_{\theta,\delta}(f; \beta) = \beta + \frac{1}{\delta}E(V_f(X) - \beta)_-$. From Theorem 2.6, under Assumption 3.1, $V_f(x) = \frac{1}{2}f(x)\{\theta - \frac{\tau(x)}{2}\} + \frac{1}{2}\{\theta + (1 - 2\theta)\frac{\tau(x)}{2}\}$ is a Lipschitz function in terms of $f$ with Lipschitz constant $c = \max_{x \in \mathcal{X}} \frac{1}{2\delta}|\theta - \frac{\tau(x)}{2}|$. Therefore, with probability at least $1 - \epsilon$, $\forall \beta \in \mathbb{R}$,

$$g_{\theta,\delta}(f; \beta) \le \hat{l}_{\theta,\delta}(f; \beta) + 2c\mathfrak{R}_n(\mathcal{F}) + 5c\sqrt{\frac{2\log\frac{8}{\epsilon}}{n}}$$

with $\hat{g}_{\theta,\delta}(f) = \beta + \frac{1}{n(1-\delta)} \sum_{i=1}^{n} (V_f(X_i) - \beta)_-$. Take $\beta = \beta^*$ and recall that $L_{\theta,\delta}(f) = \sup_{\beta \in \mathbb{R}} g_{\theta,\delta}(f;\beta)$, we have

$$L_{\theta,\delta}(f) = g_{\theta,\delta}(f;\beta^*) \le \hat{g}_{\theta,\delta}(f;\beta^*) + 2c\mathfrak{R}_n(\mathcal{F}) + 5c\sqrt{\frac{2\log\frac{8}{\epsilon}}{n}}$$

$$\le \sup_{\beta \in \mathbb{R}} \hat{g}_{\theta,\delta}(f;\beta) + 2c\mathfrak{R}_n(\mathcal{F}) + 5c\sqrt{\frac{2\log\frac{8}{\epsilon}}{n}}$$

$$= \hat{L}_{\theta,\delta}^{\text{naive}}(f) + 2c\mathfrak{R}_n(\mathcal{F}) + 5c\sqrt{\frac{2\log\frac{8}{\epsilon}}{n}}.$$

The proof is done.

# E. Additional Experimental Details and Results

### E.1. Datasets and Preprocessing.

To measure the effectiveness of the proposed methods, we conduct extensive experiments based on two classical real-world datasets, **Twins** (Almond et al., 2005) and **Jobs** (LaLonde, 1986). The Twins dataset seeks to explore the risk of low-birth weight on the mortality of the baby. It consists of 11,984 pairs of twins born in the USA between 1989 and 1991, including 50 covariates for the twin pair such as mother and father age and education, health complications and so on. Following Louizos et al. (2017), treatment $T = 1$ indicates that the baby is heavier than his/her cousin, and the outcome is the mortality for the twins. For each pair of twins, we randomly choose which baby we can observe with equal probability. The Jobs dataset is based on the National Supported Work program (Smith & Todd, 2005), in which the treatment is job training and the outcome is employment status and income (Shalit et al., 2017). The dataset consists of 3212 units (9% treated, 91% control), with 15 covariates measuring the attributes of job seekers. We generate the counterfactual outcomes on Jobs following (Wu et al., 2025), and substitute $Y(1)$ with $Y^{\text{new}}(1) = \max\{Y(1), Y(-1)\}$ to ensure monotonicity (Assumption 3.1). We denote the transformed datasets as **Twins_mono** and **Jobs_mono**. Since the counterfactual outcomes are known in both datasets, the true treatment responder indicator $R$ is known, enabling regret computation. With the main results evaluated on transformed dataset, we also evaluate the performances of our method on the original dataset when monotonicity violates in ablation studies. Training/validation/testing sets are split at ratio of $63/27/10$ in Twins and $80/10/10$ in Jobs.

### E.2. Additional Details on Configurations

**Configuration Details.** All experiments were conducted on a Core i5-1155G7 Laptop CPU. The batch size is set as 300, and training iteration set as 1000 for all models. The optimal learning rate is set as 0.0005 for TRECA. We employ the Adaptive Moment Estimation (Adam) optimizer during model training. In the TRECA model, the representation network consists of two layers, the propensity network contains three layers, and the outcome network includes two layers. All hidden layers are set to a dimensionality of 16.

### E.3. Experiments on Neural Network Depth

To examine whether impact of the space complexity in hypothesis class of predictors $\mathcal{F}$ on model performance, we conduct an additional set of experiments to evaluate the performance of our method under different layers of predictor networks that correspond to different levels of space complexity. We vary the number of layers in the predictor network and evaluate architectures with 4, 10, 30, and 50 layers under the settings of $\theta = 0.5$ and an abstention rate of $\delta = 0.1$ on the Twins_mono dataset. We also compare the performance with the predictor trained from logistic regression, i.e., $\mathbb{P}(f(x) = 1) = logit(x^\top \alpha)$. The results are summarized in Table 6.

The results indicate that increasing the depth of the predictor network consistently improves model performance. Specifically, the regret on the retained samples (**f-regret-r**) exhibits a slight but steady decrease as the depth grows from 4 to 50 layers, together with improvements on the regret on abstained samples (**f-regret-a**) and the overall samples (**f-regret-o**). The **f-regret**s using neural network is generally smaller than those using logistic regression. These findings suggest that deeper predictor networks can better capture the uncertainty structure and further enhance the effectiveness of the abstention mechanism.

*Table 6.* Performance of $f$ under different layer settings.

| layer of $f$ | 4 | 10 | 30 | 50 | logistic |
|---|---|---|---|---|---|
| **f-regret-r** | 0.0035 | 0.0035 | 0.0034 | **0.0032** | 0.0547 |
| **f-regret-a** | 0.3485 | 0.2273 | 0.0758 | 0.0530 | 0.2930 |
| **f-regret-o** | 0.0388 | 0.0264 | 0.0109 | 0.0031 | 0.1308 |

## E.4. Experiments on Time and Space Complexity

To assess and compare the computational efficiency of different models, we conducted experiments measuring their running time under identical conditions. All methods were trained on the `Twins_mono` dataset using the same experimental settings ($\theta = 0.5$, abstention rate $\delta = 0.3$) and the same hardware configuration to ensure strict comparability. For each method, we recorded the total time required to complete 1000 training epochs. The results are summarized in Table 7.

*Table 7.* Running time and memory comparison across different models.

| Model | Time | Rate |
|---|---|---|
| DragonNet | 4.30s | 10.19% |
| DESCN | 5.18s | 12.28% |
| ESCFR | 8.97s | 21.27% |
| CFRISW | 12.64s | 29.97% |
| **TRECA (ours)** | **42.18s** | **100%** |
| CFRNet | 86.83s | 205.86% |
| DeRCFR | 100.89s | 239.19% |
| CTRL | 134.20s | 318.16% |
| CEVAE | 183.75s | 435.63% |

(a) Running time comparison.

| Model | Memory(standardized) |
|---|---|
| CFRNet | 0.5 |
| DragonNet | 0.7 |
| CFRISW | 0.7 |
| **TRECA (ours)** | **1.0** |
| ESCFR | 1.3 |
| CTRL | 1.4 |
| DESCN | 1.8 |
| DeRCFR | 1.8 |
| CEVAE | 2.5 |

(b) Memory usage comparison.

The results indicate that simpler meta-learning methods such as DragonNet and DESCN achieve the shortest training times, while more complex or deep generative models such as CEVAE and CTRL require substantially longer computation. Taking TRECA as a reference point (100%), we ranked all other methods by their relative running-time ratios. As shown in the table, our proposed model TRECA lies in the middle of the spectrum. Moreover, the models that train faster than TRECA consistently exhibit larger f-regret, indicating that their efficiency comes at the cost of reduced predictive accuracy. In contrast, TRECA maintains moderate computational complexity while achieving strong predictive performance. This demonstrates that TRECA provides a favorable balance between efficiency and accuracy.

For space complexity, we record and compare the system memory training our methods and baseline methods. For standardization, we set the memory for our method as 1. Table 7 shows that our method has a medium level of memory occupation. In particular, the required memory is less than that of CTRL, another treatment responder classification method using joint learning.

