# OpenReview forum: "Treatment Responder Classification with Abstention"
_ICML.cc/2026/Conference — ICML 2026 spotlight_

### Official Review · Reviewer_iZat · 2026-02-25

**Soundness:** 3
**Presentation:** 3
**Significance:** 2
**Originality:** 2
**Overall Recommendation:** 5
**Confidence:** 3

**Summary:**

The authors extend the use of reject option to the framework of treatment responder classification. They consider a setting in which the rejection rate is controlled while minimizing an appropriate notion of classification risk under rejection.
Theorem 2.5 shows that the optimal rejection rule can be characterized as a thresholding of the conditional for uncertainty measurement (see Eq. (2)). To construct the algorithm, the authors establish a connection between the misclassification risk and CVaR in Proposition 2.6. They further provide theoretical guarantees on the consistency of the empirical risk toward the population risk in Theorem 3.3 under a monotonicity assumption of the potential outcomes (Assumption 3.1), and in Theorem 3.5 when this assumption does not hold. Finally, Theorem 3.6 establishes a control of the rejection rate.
An extensive numerical study is conducted to illustrate the empirical performance of the method.

**Compliance With Llm Reviewing Policy:**

Affirmed.

**Final Justification:**

Thank you for the discussion.

Please incorporate the relevant parts of our exchange into the revised version, and address the issues I pointed out in the literature review.

In my point of view, the notion of “selective learning” does not seem appropriate in the context of this work, and its inclusion may create unnecessary confusion with the established concept of learning with a rejection option. It may therefore be preferable to omit it.

**Key Questions For Authors:**

Beyond the points raised above, I have the following questions:
1. Is it possible to achieve rate of convergence on the excess-risk ?
2. In line 214 (second column), the authors define the augmented loss $l_{\theta , \delta}$ that includes a last term depends on $T$. While the other components of $l_{\theta , \delta}$ are quite clear from Propositions 2.6 and 3.2, the role of this additional term is less clear. It appears to be motivated by robustness considerations, but its practical and theoretical effect is not entirely transparent. Could the authors elaborate more on this point?
3. The appendix provides a generalization bound in Theorem D.2. Could the authors provide examples of settings where $c$ is bounded?

Moreover, the bound depends on $\hat{L}^{naive}$ which would itself require control.

**Limitations:**

yes

**Strengths And Weaknesses:**

1) Soundness : Most of the theoretical results are well formalized and the assumptions are clearly stated. The proofs appear to be correct (apart from minor typos). The numerical study could be further strengthened, but it provides a reasonably comprehensive comparison with existing methods on two real datasets.
I also appreciate the effort made by the authors to enable comparisons with methods that do not explicitly incorporate rejection (by rejecting high-risk samples). Although this procedure is somewhat heuristic, it allows for a meaningful empirical comparison.

However, I have several concerns :
- The theoretical analysis does not explicitly take into account the estimation of the nuisance parameters $\pi$, $\tau$, and $\mu$. Most results focus on consistency of the empirical risk for a deterministic (fixed) classifier $f$. It would be valuable to study the excess-risk of the learned estimator, rather than only consistency at a fixed $f$. This issue is only partially addressed in Theorem 3.6, which focuses on rejection rate control.

In addition, the rejection rate control established in Theorem 3.6 is not illustrated in the numerical section. Empirical validation of this theoretical result would strengthen the contribution.

- The literature review could be clarified. For instance:

Cortes et al (2016) do not formulate rejection rate control as an explicit constraint; their work is closer in spirit to Herbei and Wegkamps. (2006).

The formulation of classification with controlled rejection rate originates in Chow (1957 and 1970).

From a statistical perspective, this framework has been further studied by Denis and Hebiri (2020).

Clarifying these distinctions would improve the positioning of the contribution.

- Data splitting mechanism is not entirely clear.

Indeed, on the one hand, Algorithms 1 and 2 do not explicitly describe the estimation of $f$. It is therefore unclear whether the same data are used to estimate $\hat{\pi}$, $\hat{\tau}$, $\hat{\mu}$, and $\hat{f}$ and to calibrate the rejection rule.

Line 730 in Algorithm 1 is unclear: does $V_{\hat{f})$ rely on population quantities (as suggested by Proposition 2.6), or is the expectation replaced by an empirical average?

On the other hand, in Theorem 3.6, the selector $\hat{r}_M$ is not explicitly defined.

Moreover, the estimation of $U_{\hat{f}}$ — which appears central for understanding the impact of estimation on rejection rate control (see line 949) — is not clearly discussed. Typically, such quantities require an additional data-splitting or calibration step. Clarifying this aspect would be helpful.

- Regarding the numerical studies, I would like to raise several issues:

The number of repetitions (five) appears rather small. If computational constraints are the limiting factor, this could be mentioned explicitly.

The rejection rate itself is not analyzed in detail, although it is a central aspect of the method.


Typos :
- Lines 85–87 (first column): incomplete sentence.
- Line 194 (second column) : extra $f$.
- Line 225 (first column): expectation misplaced.
- The sentence immediately following Theorem 3.3 is unclear.
- Line 873 : the exponent $q-1$ appears to be missing in $\Vert V_f(X;\tau) -  V_f(X; \hat{\tau}) \Vert_q$.
- Line 1054 : Proposition 2 likely refers to Proposition 3.2.
- Lines 724–726 refer to  $f$ rather than $\hat{f}$.

Minor remarks:
- The authors introduce the f-regret as a convex combination of false positive and false negative rates. However, throughout the paper, $\theta = 1/2$ which reduces the f-regret to the misclassification risk. It would be helpful to clarify the motivation for introducing the more general notion.
- Figure 2 is difficult to read due to its size. It should also be specified whether the reported f-regret is computed in-sample or out-of-sample.
- The proof of Theorem 3.5 should be placed before the proof of Theorem 3.6 (permute Appendices D6 and D7).




2) Presentation: The paper is generally well written (aside from the points mentioned above), and the technical tools are clearly introduced. As a non-expert in causal learning, I found the formalization of treatment responder classification particularly helpful and pedagogical.



3) Significance: While I am not an expert in causal learning, I believe the problem addressed is important. Incorporating a reject option into treatment responder classification may have significant practical implications.
That said, I remain somewhat unconvinced that the reject option is fully developed in both theory and practice. In particular, although the rejection rule is built using the risk $V_f$, the theoretical and empirical control of the rejection rate could be explored more thoroughly. Furthermore, Assumption 2.4, which requires continuity of the random variable $V_f (X)$, may be restrictive; additional discussion and illustrative examples would be beneficial.



4) Originality: To my knowledge, introducing a reject option into treatment responder classification appears to be novel. From this perspective, the contribution is original.
However, the overall impact could be strengthened by a more detailed theoretical and empirical study of the reject option component itself, particularly regarding excess-risk and rejection calibration.

---

> ### Author Rebuttal · Authors · 2026-03-31
>
> We would like to express our genuine gratefulness for your thorough reading as well as your acknowledgement on the series of theoretical results. The questions and typos you mentioned substantially help us increase the clearance of the paper, and will be carefully revised in the camera-ready version. The detailed answer to the key questions are summarized as follows.
>
> > Q1. Is it possible to achieve rate of convergence on the excess-risk ?
>
> This is indeed a good question. Following the idea of Cortes et al. (2023), excessive risk $\tilde L_{\theta,\delta}(f)-L_{\theta,\delta}(f)$ measures the gap between targeted and surrogate loss. We develop the following asymptotic theory on the excessive risk.
> > Suppose the assumptions of Theorem 3.3 hold. Let $\tilde L_{\theta,\delta}(f)=\sup_{\beta\in\mathcal R}
> E_{\mathcal D}\\!\left[\tilde l_{\theta,\delta}(X,T,Y,f;\hat\tau,\hat\mu,\hat\pi,\beta)\right]$
> be the empirical surrogate loss, where the indicator term $I\\{V_f(X;\tau)\le \beta\\}$ in $l_{\theta,\delta}$ is replaced by $\alpha\\,\phi(V_f(X)-\beta)$, with $\alpha>0$ and $\phi$ a surrogate function. Assume further that: (a) $\phi$ is Lipschitz; (b) The empirical Rademacher complexity of $\mathcal F$ satisfies$\mathcal R_n(\mathcal F)=O(n^{-1/2}).$ Then, $$
> \bigl|L_{\theta,\delta}(f)-\tilde L_{\theta,\delta}(f)\bigr|=O_p(n^{-1/2}).$$ That is, the empirical excessive risk converges at order $n^{-1/2}$.
>
> The proof decomposes the error as
> $$|L_{\theta,\delta}(f)-\tilde L_{\theta,\delta}(f)|\le|L_{\theta,\delta}(f)-\bar L_{\theta,\delta}(f)|+|\bar L_{\theta,\delta}(f)-\tilde L_{\theta,\delta}(f)|,$$
> where $\bar L_{\theta,\delta}(f)=\sup_{\beta\in\mathcal{R}}\{\alpha\phi(V_f(X)-\beta)(V_f(X)-\beta)\}$ is the population surrogate loss. The first term is $O_p(n^{-1/2})$ by Rademacher complexity argument, and the second term is also $O_p(n^{-1/2})$ by similar arguments as in Theorem 3.3. Combining the two bounds proves the result. The proof details will be added in the camera-ready version.
>
> > Q2. Could the authors elaborate more on the additional term in $l_{\theta,\delta}$?
>
> Thank you for pointing this out. The last term is indeed included for robustness and debiasing consideration. More concretely, under monotonicity the CVaR objective depends on $V_f(X;\tau)$, and hence on the nuisance quantity $\tau(X)$. A naive plug-in estimator can be quite sensitive to bias in $\hat{\tau}$ and also to the optimization over $\beta$. As a classical idea in double machine learning, the robust estimator appends an addition term built from $(\mu,\pi)$. Its expectation is zero when either the propensity model $\pi$ or the outcome model $\mu$ is correctly specified, so it corrects first-order bias in the plug-in loss without shifting the target itself.
>
> > Q3. The appendix provides a generalization bound in Theorem D.2. Could the authors provide examples of settings where $c$ is bounded?
>
> In Theorem D.2, $c=\max_x\frac{1}{2\delta}|\theta-\tau(x)/2|$ is naturally bounded observing that $|\tau(x)|\le 2$ as $Y\in\\{-1,1\\}$ in treatment responder classfication setting. The $\hat{L}_{\theta,\delta}^{naive}$ is bounded for the same reason.
>
> **Reponse to other remarks**
>
> > Q4. Motivation of introducing $\theta$.
>
> Thanks for the question. Our motivation for introducing $\theta$ is that the false negative and false positive samples may be weighted differently in practice, thus to broaden the application scenario of the paper. We have supplemented experiments on cases when $\theta\neq 0.5,$ with details provided in the reply to Reviewer `QoyP`.
>
> > Q5. Continuity of $V_f(x)$.
>
> Assumption 2.4 is satisfied for general covariates, e.g., when $X=(X_1,X_2)$, where $X_1$ are discrete random variables with finite classes, $X_2$ are continuous variables with bounded density. The conditional risk $V_f(X)$ is a differentiable function in terms of $X_2,$ and thus has continuous density function.
>
> > Q6. Data splitting mechanism.
>
> In this paper, we choose the nuisance parameters on the entire training dataset to enhance time efficiency. The details on our algorithm have been provided in Algorithm 1.
>
> > Q7. Further empirical study on the reject-option component and different repetitions.
>
> Thanks for your concern. We have added the following sensitivity analysis on the abstention rate $\delta$, which is the critical parameter controlling the prevalence of rejectiom-option in both Twins_mono (top) and Jobs_mono (bottom) datasets. Please refer to Reviewer `k8Nf` for comparison between the results under different repetitions.
>
> |$\delta$|TRECA(w/o mono.)|TRECA|CFRNet|DragonNet|
> |-|-:|-:|-:|-:|
> |0.1|0.010|0.004|0.030|0.028|
> |0.2|0.020|0.005|0.013|0.038|
> |0.3|0.014|0.005|0.022|0.020|
> |0.4|0.010|**0.003**|0.018|0.050|
> |0.5|**0.003**|0.007|0.029|0.042|
>
> |$\delta$|TRECA(w/o mono.)|TRECA|CFRNet|DragonNet|
> |-|-:|-:|-:|-:|
> |0.1|0.062|0.104|0.048|0.252|
> |0.2|0.066|0.083|0.052|0.209|
> |0.3|0.033|0.018|0.043|0.087|
> |0.4|0.010|0.004|0.016|0.014|
> |0.5|0.004|0.006|0.028|0.041|

---

> > ### Author Rebuttal · Reviewer_iZat · 2026-04-03
> >
> > Thank you for your clarification.
> >
> > However, several points remain unclear to me.
> >
> > **Regarding Q5**, it would be helpful to be more concrete by specifying an explicit distribution for $X$ and carrying out the corresponding computations.
> >
> > **Regarding Q1 and Q6**, it seems that you use the entire training dataset to calibrate the nuisance parameter. Do I understand correctly that the same dataset is also used to calibrate the rejection rate of the classifier --- in other words, to estimate $F_{V_f}^{-1}$? If so, I am not fully convinced that this would yield a reliable control of the excess-risk (by excess-risk, I mean the excess-risk of $\hat{f}$).
> >
> > **Regarding Q7**, while I acknowledge the additional experimental results, I would expect a clearer evaluation of the rejection rate. More precisely, it would be interesting to enforce a target rejection rate $\delta$ and verify that the numerical results match this $\delta$, up to estimation error.
> >
> > For now, I will keep my score unchanged, but I may revise it after further discussion with the other reviewers.

---

> > > ### Author Response · Authors · 2026-04-04
> > >
> > > Thanks for your careful reading and constructive suggestions! We would like to provide following responses on your valuable follow-up questions:
> > >
> > > > Q5.  It would be helpful to be more concrete by specifying an explicit distribution for $X$ and carrying out the corresponding computations.
> > >
> > > Below we provide a concrete example when Assumption 2.4 holds which was briefly described due to space limitation.
> > >   * **Example:** For simplicity, let $\theta=1/2,$ and consider the following setting
> > >
> > >   |Parameter|Condition|Example|
> > >   |-|-|-|
> > >   |$X$|One-dimensional continuous covariate taking value in $[-1,1]$|$Unif(-1,1)$|
> > >   |$S(x):=\frac{1-\tau(x)}{2}$|Continuous differentiable **($C^1$)** and monotonically increasing w.r.t $x$, $S(0)=0$ and $S(-1)+S(1)=0$|$S(x)=Sigmoid(x)$|
> > >   |$f$|Thresholded rule $f(x)=-1$ if $x\le 0$ and $f(x)=1$ if $x>0$.|N/A|
> > > ||
> > >
> > > From Proposition 3.2, $V_f(x)$ can be identified as $V_f(x)=\frac{1}{2}f(x)S(x)+\frac{1}{4}.$ Since $S\in C^1,$ $S(X)$ is a continuous differentiable function of a continuous random variable and its density $p_S\in C^1.$ From the monotonicity of $\tau$ and the fact that $S(0)=0,$ after derivation the density $p\_Z$ of $Z=f(X)\frac{1-\tau(X)}{2}$ satisfies $p_Z(z)=p_S(z)+p_S(-z)$ for $z\in (0,S(1))$ and is also $C^1$, hence Assumption 2.4 is satisfied.
> > >   * **Relaxation of Assumption 2.4:** Meanwhile, we would like to clarify that the main role of Assumption 2.4 is to provide a well-defined quantile $F_{V_f}^{-1}(\delta).$ Therefore Assumption 2.4 can be relaxed by defining the quantile as $F_{V_f}^{-1}(\delta)=\inf\\{x:F_{V_f}(x)\ge\delta\\}.$ Extending the definition of quantiles to relax the continuity assumption is well-studied in [1], and therefore **not** addressed as a key challenge in the original paper. We will add extended discussion on this assumption in the revised manuscript.
> > >
> > > [1] Kallus N. Treatment effect risk: Bounds and inference. Management Science, 2023.
> > >
> > > > Q1 and Q6. Is the same dataset is also used to calibrate the rejection rate of the classifier --- in other words, to estimate $F_{V_f}^{-1}$?
> > >
> > > * Thanks for the great question! The same training data are used to learn the abstention threshold, but importantly our method does **not** estimate $F_{V_f}^{-1}(1-\delta)$ through a separate calibration step. Instead, we optimize the loss $L_{\theta,\delta}(f)$ on retained samples directly. From Proposition 2.6, the threshold can be expressed as $$F\_{V\_f}^{-1}(\delta)=\arg\max_{\beta\in\mathbb{R}}  \\{\beta+\frac{1}{1-\delta} E(V\_f(X)-\beta)\_-\\},$$ and the optimal abstention rule is given by thresholding the conditional risk on the maximum. Therefore, rather than first estimating the CDF $F_{V_f}$ and then plugging in its quantile, we learn the threshold **jointly** with the predictor through in loss optimization.
> > >
> > >  * The main advantage of this **joint-training** procedure is that we avoid an extra CDF estimation step, which would introduce additional plug-in error and can be less efficient. This enables us to derive efficient convergence of the estimated loss at the rate of $n^{-1/2}$ in Theorem 3.3. We will revise the manuscript to clarify this point explicitly.
> > >
> > > > Q1 and Q6. Why this would yield a reliable control of the excess-risk of $\hat f$.
> > >
> > > While we totally agree that controlling the excess-risk of the learned predictor $\hat f,$ $L_{\theta,\delta}(\hat f)-L_{\theta,\delta}(f^\ast)$ is an important question, we would like to remark that this is a general challenge in policy learning. As Theorem 3.3 of our work establishes bound on estimated loss, it is generally hard to control the excess-risk of $\hat f$ since small error on loss function does not imply small distance between minimizers. We will address this as a future work requiring fundamental extension on policy-learning theory.
> > >
> > > > Q7. It would be interesting to enforce a target rejection rate and verify that the numerical results match this, up to estimation error.
> > >
> > > We follow this suggestion to add experiments reporting the **rejection gap** as the absolute difference between the realized rejection rate and the target $\delta$, on the **Jobs\_mono** dataset for our method TRECA and two variants, each evaluated on both Within-samples (W) and Out-of-samples (O). **Results are timed by $10^4$.**
> > >
> > > |$\delta$|TRECA (W)|TRECA (O)| w/o mono (W)| w/o mono (O)| w/o uncf. (W)| w/o uncf. (O)|
> > > |-|-:|-:|-:|-:|-:|-:|
> > > |0.1|3|25|3|25|4|27|
> > > |0.2|2|19|2|20|2|19|
> > > |0.3|0|12|0|13|1|13|
> > > |0.4|1|6|1|6|1|6|
> > > |0.5|2|0|2|1|2|0|
> > > |0.6|1|6|1|6|1|6|
> > > |0.7|1|12 |0|12|1|12|
> > > |0.8|2|19|2|19|2|19|
> > > |0.9|3|25|3|25|3|25|
> > > ||
> > >
> > > The results show that the rejection gap is uniformly small across with at a maximum of 0.0027, showing **realized rejection rate closely tracks the target $\delta$, up to a very small estimation error**.
> > >
> > > ***
> > >
> > > We would highly appreciate it if you could take into account our response when updating the final rating and having discussions with AC and other reviewers.
> > >
> > > Thanks for your time,
> > >
> > > Authors of # 27667

---

### Official Review · Reviewer_k8Nf · 2026-03-05

**Soundness:** 3
**Presentation:** 3
**Significance:** 3
**Originality:** 3
**Overall Recommendation:** 5
**Confidence:** 3

**Summary:**

The authors propose an interesting topic by introducing an abstention option into treatment responder classification. They reveal the implicit relationship between causal misclassification risk with abstention and Conditional Value at Risk (CVaR), and propose the TRECA framework to learn the classification rule under loose convergence conditions.

**Compliance With Llm Reviewing Policy:**

Affirmed.

**Key Questions For Authors:**

Mentioned in Weaknesses part.

**Limitations:**

yes

**Strengths And Weaknesses:**

# Strengths:
1.	The overall writing is excellent. Although the task itself is relatively simple, the paper provides thorough validation, from both theory and experiments, on how to handle the problem after introducing abstention, and explains the proposed method in a clear, step-by-step manner.
2.	The theoretical analysis is comprehensive and well explained, offering insights and interpretations from multiple perspectives.
3.	The method achieves very strong performance on mainstream benchmark datasets.

# Weaknesses:
1.	Although the topic is interesting, the Section Introduction contains too much task description, especially in the first three paragraphs, and does not clearly articulate what new challenges arise and what needs to be solved after introducing abstention.
2.	The method itself does not provide a detailed description of the network architecture.
3.	The proposed method involves several hyperparameters, but the paper lacks analyses on how these hyperparameters affect performance as well as ablation studies. Adding them would further validate the advantages of the framework.

---

> ### Author Rebuttal · Authors · 2026-03-31
>
> Thanks for your recognition on the writing, theoretical development and experimental results of our work! The responses to your questions are provided as follows.
>
> > Q1. Introduction and problem framing.
>
> Thanks for your valuable suggestion. The introduction will be carefully reorganized in the revised manuscript. The revised version will separate background from the actual technical challenges introduced by abstention: identification of responder risk with latent counterfactual labels, learning under an abstention-rate constraint, orthogonal estimation with nuisance parameters, and partial-identification extensions when monotonicity / unconfoundedness are relaxed.
>
> > Q2. Network architecture details.
>
> Thanks for your concern. We have summarized a table on the key configuration details and network structure features summarized as follows. Please kindly check further details on the network structure and configurations in Appendix E.2.
>
> |Parameter|Value|
> |-|-|
> |CPU|Intel Core i5-1155G7|
> |batch size|500|
> |epoch|1000|
> |learning rate|0.0005|
> |Representation network hidden layer|2|
> |propensity network hiddenlayer|3|
> |outcome network hidden layer|2|
> |representation network hidden layer|2|
> |hidden dimension|16|
> |activation for hidden layer|ELU|
> |activation for output layer|Sigmoid|
> |optimizer|Adam|
>
> >Q3. Further evaluation on the impact of hyperparameters.
>
> In response to your valuable concern, we have added sensitivity experiments on three important hyperparameters, the learning rate, network depth for predictor and abstention rate $\delta.$ We also added $\theta\neq 0.5$(refer to QoyP-Q4) experiments and the STAR benchmark(refer to QoyP-Q2). These additions are intended to make the dependence on key design choices much more transparent.
> The results are as follows.
>
> | Learning rate |Test f-regret-r | Test f_regret-a | Test f-regret-o | Train f-regret-r | Train f_regret-a | Train f_regret-o |
> |-------|---------------|-------------------|------------------------|----------------|--------------------|------------------------|
> | 0.0001|0.008 ± 0.010 | 0.280 ± 0.211     | 0.035 ± 0.026          | 0.011 ± 0.014  | 0.284 ± 0.216      | 0.039 ± 0.029    |
> | 0.0005 | 0.001 ± 0.002 | 0.352 ± 0.171     | 0.039 ± 0.017          | 0.003 ± 0.003  | 0.322 ± 0.166      | 0.036 ± 0.017  |
> |0.001 | 0.003 ± 0.001 | 0.263 ± 0.157     | 0.029 ± 0.015          | 0.004 ± 0.002  | 0.251 ± 0.150      | 0.029 ± 0.015 |
>
> | layer of *f* | 4 | 10 | 30 | 50 |
> |---|---:|---:|---:|---:|
> | f-regret-r | 0.0035 | 0.0035 | 0.0034 | **0.0032** |
> | f-regret-a | 0.3485 | 0.2273 | 0.0758 | 0.0530 |
> | f-regret-o | 0.0388 | 0.0264 | 0.0109 | 0.0031 |
>
> |$\delta$|TRECA+|TRECA|CFRNet|DragonNet|
> |-|-:|-:|-:|-:|
> |0.1|0.010|**0.004**|0.030|0.028|
> |0.2|0.020|**0.005**|0.013|0.038|
> |0.3|0.014|**0.005**|0.022|0.020|
> |0.4|0.010|**0.003**|0.018|0.050|
> |0.5|**0.003**|0.007|0.029|0.042|
>
> ---
>
> We present the experimental results under different repetitions of ``iZat`` here.
>
> | Exps  | Test f-regret-r| Test f-regret-a| Test f-regret-o| Train f-regret-r | Train f-regret-a | Train f-regret-o| Total Running Time (s) |
> |-|-|-|-|-|-|-|-
> |20|.005±.002|.306±.183|.035±.019|.006±.005|.297±.184|.035±.020|1437.25|
> |50|.004±.002|.305±.183|.035±.019|.005±.005|.290±.181|.035±.020|3105.47|

---

> > ### Author Rebuttal · Reviewer_k8Nf · 2026-04-04
> >
> > The author's response addressed my concerns, so I decided to maintain my positive rating.

---

> > > ### Author Response · Authors · 2026-04-07
> > >
> > > Dear Reviewer k8Nf,
> > >
> > > We really appreciate that our response adequately addressed your concerns. Many thanks for maintaining your positive rating of 5.
> > >
> > > Thanks for your time,
> > >
> > > The authors of #27667

---

### Official Review · Reviewer_QoyP · 2026-03-13

**Soundness:** 3
**Presentation:** 3
**Significance:** 4
**Originality:** 3
**Overall Recommendation:** 5
**Confidence:** 3

**Summary:**

The paper presents a new method for treatment responder classification under abstention. Specifically, they introduce TRECA to learn (1) a treatment responder classifier and (2) an abstention rule. They provide a neat theoretical framework built on results from optimization and causal inference, notably connecting misclassification risk under abstention to Conditional Value at Risk (CVaR), and test the method on two real-world datasets (Twins and Jobs).

**Compliance With Llm Reviewing Policy:**

Affirmed.

**Final Justification:**

My justification is the same. The authors answer the questions I had.

**Key Questions For Authors:**

1. While Figure 2 examines performance across different abstention rates \delta, it remains unclear for me how sensitive the method is to this choice in practice. More importantly, how should a practitioner select \delta in a real-world deployment, where one cannot simply cross-validate on labeled responder outcomes?
2. The paper fixes \theta = 0.5 throughout all experiments. Is this choice principled or simply a default? This is worth discussing, because in practice tuning θ is non-trivial in the causal setting, since the true responder label R depends on counterfactual outcomes that are never jointly observed, standard label-based tuning procedures (e.g., optimizing recall via cross-validation) are not directly applicable. This is particularly relevant in clinical contexts where one may strongly prefer to minimize false negatives.

**Limitations:**

Yes.

**Strengths And Weaknesses:**

The paper is technically sound with clean theoretical results. The clinical motivation is on point, and the constraint-based formulation in Equation (1) along with its explanation is neat. On that note, I was wondering whether this constraint formulation is standard in the literature or an original contribution of the authors, either way, appropriate references should be added, or it should be explicitly credited as a contribution.
The experiments are somewhat limited. Jobs is a small dataset, and Twins may carry inherent biases. I would recommend adding the STAR (Kallus et al., 2018) and ACTG datasets for completeness and broader empirical validation.
The abstract could be rewritten more clearly. As a reader unfamiliar with abstention, I found it difficult to grasp the problem from the abstract alone, the key explanatory sentence is ambiguous, and I only fully understood the setup after reading the introduction and background sections.
Finally, there are minor writing issues: a typo in the first sentence of the intro ("are" should be "is"), and I think there is a missing word in the sentence immediately preceding Equation (1) "solve".

---

> ### Author Rebuttal · Authors · 2026-03-31
>
> Thanks for your valuable questions! The detailed answers are provided below.
>
> >Q1. **Controlled-abstention formulation.**
>
> Thanks for your concenr. The main contribution of our paper is bringing this formulation to **treatment responder classification**, identifying the corresponding causal risk, and connecting it to CVaR. The revision will add the appropriate references on controlled rejection and state the contribution at the right level of granularity.
>
> >Q2. **Broader experiments.**
>
> The rebuttal adds experiments on the STAR dataset, in addition to Twins/Jobs. This expands the empirical section beyond the two benchmarks in the submission. ACTG is not added in this round; STAR was prioritized as the third benchmark to address the request for broader validation. The results on STAR dataset are as follows.
>
> |Model|Test f-REGRET| Test f_REGRET_abs|Test f_REGRET_overall|Train f-REGRET|Train f_REGRET_abs|Train f_REGRET_overall|
> |---|---|---|---|---|---|---|
> |**TRECA**| **0.177 ± 0.104** | 0.234 ± 0.096 | 0.183 ± 0.101 | **0.180 ± 0.102** | 0.239 ± 0.100 | 0.186 ± 0.099 |
> | DragonNet | 0.365 ± 0.030 | 0.366 ± 0.030 | 0.366 ± 0.030 | 0.363 ± 0.028 | 0.373 ± 0.032 | 0.364 ± 0.028 |
> | DESCN | 0.194 ± 0.063 | 0.193 ± 0.097 | 0.194 ± 0.065 | 0.196 ± 0.065 | 0.205 ± 0.096 | 0.197 ± 0.066 |
> | DeRCFR | 0.366 ± 0.045 | 0.368 ± 0.051 | 0.366 ± 0.045 | 0.364 ± 0.044 | 0.367 ± 0.049 | 0.364 ± 0.045 |
>
> >Q3. **How to choose $\delta$ in practice?**
>
> $\delta$ is meant to encode deployment capacity, review budget, or the maximum fraction of cases that can be sent for further investigation. It is therefore not a hyperparameter that must be cross-validated on latent responder labels. In practice, one chooses $\delta$ from operational constraints and then reads the resulting risk-coverage trade-off. We will make this interpretation explicit in the revised manuscript.
>
> >Q4. **Why $\theta=0.5$ in the main experiments?**
>
> $\theta=0.5$ is a symmetric default matching the standard f-regret used in responder-classification work. It is not intrinsic to TRECA. The rebuttal adds experiments with $\theta\neq 0.5$ to make this explicit. In deployment, $\theta$ should be chosen from the relative cost of false positives and false negatives; in clinical settings this naturally allows prioritizing false-negative control when safety is the main concern.
>
> | $\theta$ | Test f-regret-r | Test f-regret-a | Test f-regret-o | Train f-regret-r | Train f-regret-a | Train f-regret-o |
> |---|---|---|---|---|---|---|
> | 0.2 | 0.010 ± 0.008 | 0.351 ± 0.200 | 0.044 ± 0.024 | 0.012 ± 0.010 | 0.352 ± 0.201 | 0.047 ± 0.027 |
> | 0.5 |  0.001 ± 0.002 | 0.352 ± 0.171 | 0.039 ± 0.017 | 0.003 ± 0.003 | 0.322 ± 0.166 | 0.036 ± 0.017 |
> | 0.8 | 0.004 ± 0.001 | 0.260 ± 0.165 | 0.029 ± 0.016 | 0.005 ± 0.003 | 0.250 ± 0.173 | 0.030 ± 0.018 |
>
> >Q5. **Abstract and presentation.**
>
> The abstract will be rewritten so that a reader unfamiliar with abstention can understand the setup from the abstract alone. The noted writing issues will also be fixed, including the typo in the first sentence of the introduction and the missing word before Eq. (1).
>
> ---
>
> We present the experimental results of ``tzbK-Q3`` here.
>
> | Model  | Test f-regret-r | Test f-regret-a | Test f-regret-o | Train f-regret-r | Train f-regret-a | Train f-regret-o |
> |--------|---------------|-------------------|------------------------|----------------|--------------------|------------------------|
> | TRECA  | **0.001 ± 0.002** | 0.352 ± 0.171     | 0.039 ± 0.017    | **0.003 ± 0.003** | 0.322 ± 0.166      | 0.036 ± 0.017          |
> | Quince[10] | 0.216 ± 0.118 | 0.055 ± 0.028     | 0.200 ± 0.106          | 0.214 ± 0.104  | 0.058 ± 0.049      | 0.198 ± 0.094          |
> | PTCATE[11] | 0.004 ± 0.001 | 0.015 ± 0.013     | 0.005 ± 0.002          | 0.004 ± 0.002  | 0.011 ± 0.017      | 0.005 ± 0.002          |

---

> > ### Author Rebuttal · Reviewer_QoyP · 2026-04-02
> >
> > Thank you for these answers! I am keeping my positive assessment of the paper.

---

> > > ### Author Response · Authors · 2026-04-02
> > >
> > > Dear Reviewer QoyP,
> > >
> > > We really appreciate your recognition of our work and your kind words, and we are happy to hear that your concerns have been adequately addressed. Again, thank you for your valuable suggestions which have undoubtedly contributed to improving the quality of our paper.
> > >
> > > Many thanks,
> > >
> > > The authors of #27667

---

### Official Review · Reviewer_tzbK · 2026-03-14

**Soundness:** 3
**Presentation:** 3
**Significance:** 3
**Originality:** 3
**Overall Recommendation:** 4
**Confidence:** 4

**Summary:**

This paper studies treatment responder classification with abstention, i.e., a causal decision-making setting in which the learner is allowed to withhold a treatment recommendation on uncertain individuals and optimize performance on the retained cases. In that sense, the paper can be viewed as extending prior work on treatment responder classification to a selective-decision setting.

**Compliance With Llm Reviewing Policy:**

Affirmed.

**Key Questions For Authors:**

1. Can you establish a lower bound, minimax benchmark, or some impossibility result for this problem? Related policy-learning work provides lower-bound benchmarks [9], and that would make the theoretical contribution easier to assess.
2. Can you compare against a simple but strong baseline that estimates CATE and abstains whenever an uncertainty interval crosses zero (or another relevant decision threshold) ?

**Limitations:**

My main limitation concern is practical relevance. In real applications, abstention is not free: it may require additional testing, delayed treatment, or escalation to a human expert. A hard constraint on the abstention rate is interpretable, but it does not by itself capture those operational costs. Unless the paper can show a clearer advantage over simpler uncertainty-aware CATE-based or direct policy-learning baselines, the practical benefit of the framework remains somewhat unclear.

**Strengths And Weaknesses:**

The formulation is clean and straightforward, and the paper tackles a natural problem. The connection to selective classification and reject-option learning is also conceptually appealing, and the conditional-risk or CVaR perspective is a technically interesting way to formalize abstention. To the best of my knowledge, this paper is among the first works to study abstention specifically in the setting of treatment responder classification.

That said, the novelty claim should be stated more carefully. Selective classification itself has a long history [1–4], and closely related causal decision settings with abstention or deferral have already appeared recently [5–7]. In particular, the authors should position the paper more explicitly against Sawarni et al. [7] (with Syrgkanis), as well as recent deferral-based causal decision papers such as Ghoummaid and Shalit [5] and Gao and Yin [6]. If the present work was developed independently, that is perfectly fine, but the distinction in scope and setting should still be clarified carefully.

My main concern is conceptual rather than purely technical. Precisely because the problem is so natural, the paper should explain more clearly why abstention is the right causal object here, rather than simply a special case of uncertainty-aware treatment-effect estimation. In practice, one strong alternative is to estimate the CATE and then treat only when the estimated effect is positive or exceeds a practically meaningful threshold; recent work explicitly notes that such thresholding is common, and that one may defer decisions when the CATE estimate is highly uncertain [10,11]. Another alternative is to optimize treatment allocation directly through policy-learning methods [8,9]. Relative to these alternatives, it is still unclear to me what the proposed formulation gains, either theoretically or practically.

Relatedly, I would like the paper to clarify whether its advantage comes from the abstention mechanism itself, from directly optimizing responder classification instead of CATE estimation, or from the special structural assumptions of the setting (e.g., binary treatment/outcome, monotonicity, and the responder-classification abstraction). Without such clarification, it is difficult to tell whether the paper identifies a genuinely new causal decision problem or mainly recasts an existing one in selective-classification language.

The empirical results are promising, but they do not fully resolve the practical question. Much of the evaluation is conducted on transformed monotone versions of semi-synthetic benchmarks, which is reasonable for controlled benchmarking but only partially representative of deployment. I appreciate that the appendix adds a unified uncertainty-based abstention procedure to the baseline CATE estimators; however, I would still like to see stronger baselines based on calibrated CATE uncertainty, interval-based deferral, or recent decision-targeted CATE methods [10,11]. Since the motivating application is treatment allocation, reporting downstream policy value in addition to responder-classification regret would also strengthen the empirical case.

1] Chow, C. “On Optimum Recognition Error and Reject Tradeoff.” IEEE Transactions on Information Theory, 16(1):41–46, 1970.

[2] Herbei, R., and Wegkamp, M. H. “Classification with Reject Option.” Canadian Journal of Statistics, 34(4):709–721, 2006.

[3] El-Yaniv, R., and Wiener, Y. “On the Foundations of Noise-free Selective Classification.” Journal of Machine Learning Research, 11:1605–1641, 2010.

[4] Geifman, Y., and El-Yaniv, R. “Selective Classification for Deep Neural Networks.” Advances in Neural Information Processing Systems, 2017.

[5] Ghoummaid, M., and Shalit, U. “When to Act and When to Ask: Policy Learning With Deferral Under Hidden Confounding.” Advances in Neural Information Processing Systems, 37, 2024.

[6] Gao, R., and Yin, M. “Confounding-Robust Deferral Policy Learning.” Proceedings of the AAAI Conference on Artificial Intelligence, 39(13):14238–14246, 2025.

[7] Sawarni, A., Jin, J., Whitehouse, J., and Syrgkanis, V. “Policy Learning with Abstention.” arXiv:2510.19672, 2025.

[8] Kitagawa, T., and Tetenov, A. “Who Should Be Treated? Empirical Welfare Maximization Methods for Treatment Choice.” Econometrica, 86(2):591–616, 2018.

[9] Athey, S., and Wager, S. “Policy Learning With Observational Data.” Econometrica, 89(1):133–161, 2021.

[10] Jesson, A., Mindermann, S., Gal, Y., and Shalit, U. “Quantifying Ignorance in Individual-Level Causal-Effect Estimates under Hidden Confounding.” Proceedings of the 38th International Conference on Machine Learning, PMLR 139, 2021.

[11] Frauen, D., Melnychuk, V., Schweisthal, J., van der Schaar, M., and Feuerriegel, S. “Treatment Effect Estimation for Optimal Decision-Making.” arXiv:2505.13092, 2025.

---

> ### Author Rebuttal · Authors · 2026-03-31
>
> Thank you for the careful reading and inspiring feedback! We have addressed each of the raised weaknesses and questions in detail below.
>
> > Q1. Further clarification on novelty and position.
>
> Thanks for your insightful comment and helpful remark on the references on classical abstention works and deferral systems. We would like to provide the following positioning of our paper in the context of abstention learning:
>
> * Comparison with classical abstention learning [1-4,7]:
>   * We study abstention in treatment responder classification, where the target is the **latent responder label** $R=I(Y(1)>Y(0))$, not a causal object that can not be observed from the samples. This causes fundamental difference in formulation, identification and theories.
>   * We show that this abstention problem admits an equivalent conditional value-at-risk (CVaR) characterization. We also provide partial-identification-based extensions for violations of monotonicity and unconfoundedness, which are specific and crucial assumptions in responder-classification settings.
>
> All of the above developments are **critical in causal scenario** which has not been fully explored by classical abstention works.
>
> * Comparison with deferral policy learning [5-6]: We appreciate the reviewer for mentioning the sounding works on deferral systems. As mentioned in the related work (Appendix A.2), we share a similar goal
> of minimizing risk under constraint on the proportion of deferred / abstained samples. However, a key difference is that while [5,6] aim at learning policy $\pi$ that maximizes outcome $EY(\pi(x))$, our work focuses on predicting $R=I(Y(1)>Y(0)).$ This involves **two key challenges**:
>   * As in causal inference we can only observe either $Y(1)$ or $Y(0)$ for each individual, $R$ can not be directly identified from observed samples, inducing non-trivial discussion on possible **identification conditions**.
>   * As $R$ is a binary reward, we manage to tackle the discontinuous indicator function $I(V_f(X)\le\beta)$ and construct **doubly robust estimator** on the classifier (predictor), which yields different double robustness property comparing with AIPW estimator on CATE and is not fully explored in [5,6].
> The comparisons are summarized as follows:
>
> |Work|Causal Object|Identification Issue|Main Theories|
> |-|-|-|-|
> |Classical abstention learning[1-4,7]|No|N/A|Excessive risk bound|
> |Deferral Systems[5,6]|Outcome $Y$|No|Consistency, Generalization bound|
> |Our Work|Responder status $R$|Yes|Double robustness, Rejection rule control, Excessive risk bound|
>
> > Q2. Can you establish a lower bound, minimax benchmark, or some impossibility result for this problem?
>
> Great thanks for this insightful suggestion! Following the notations in the main paper, we develop a theorem regarding the lower bound on regret $R(\hat{f}):=L_{\theta,\delta}(\hat{f})-L_{\theta,\delta}(f^*)$ quantifying the gap between the losses on learned and optimal classifier:
>
> > **Theorem 1'.** Let $\mathcal F$ be a predictor class with finite VC dimension $VC(\mathcal F)=d<\infty,$ where $d$ does not depend on $n$. Suppose the assumptions of Theorem 3.3 hold, and assume that $\delta d\in\mathbb Z$ and $n/d\in\mathbb Z$. Then there exists a covariate distribution $\mathcal P_X$ such that
> $$
> \liminf_{n\to\infty}
> \sqrt n\\,\inf_{\hat f}\sup_{\tau}E\\!\left[R(\hat f)\right]\ge
> 0.44\sqrt{S_{\mathcal P}\\,VC(\mathcal F)},$$
> where $S_{\mathcal P}=E_{\mathcal P}\left[\frac{\sigma(X)}{\pi(X)(1-\pi(X))}\right],\qquad\sigma^2(X)=Var[Y\mid X,T].$
>
> In particular, the minimax regret over $\mathcal F$ is lower-bounded at order $n^{-1/2}$. The proof is sketched as follows, with details provided in the revised manuscript. The proof constructs a least-favorable family of data-generating distributions on $d$ shattered cells $A_1,\dots,A_d$, with $A^0$ and $A^1$ representing cells that should be abstained and retained. The treatment effect is chosen as a local alternative, with $\tau(x)\asymp c_j/\sqrt n$ on each cell, where $c_j=m$ on $A^0$ and $c_j=3m$ on $A^1$ for some constant $m=0.75\sqrt{S_{\mathcal P}/d}$. Because each cell carries information of order $S_{\mathcal P}/d$, estimating $c_j$ has efficient asymptotic variance $S_{\mathcal P}/d$, so the probability of mistaking the sign of $c_j$ or the threshold decision $c_j\gtrless 2m$ stays bounded away from zero. Summing these cellwise errors over the $d$ shattered cells yields a total regret of order $\sqrt{S_{\mathcal P}d/n}$, which gives the lower bound $\inf_{\hat f}\sup_\tau E[R(\hat f)] \gtrsim \sqrt{\frac{S_{\mathcal P}\\,VC(\mathcal F)}{n}}.$
>
> > Q3. Comparison with CATE uncertainty and decision-targeted CATE methods is recommended.
>
> Thanks for your valuable remark! We have compared our work with the decision-targeted CATE methods [10] and [11]. Please see the result table in the reply to Reviewer `QoyP` for space limitation. Results demonstrate that our method outperforms decision-targeted CATE methods in Twins_mono datasets.

---

> > ### Author Rebuttal · Reviewer_tzbK · 2026-04-08
> >
> > Thank you for your reply. I now have a better understanding of the issues at hand.

---

> > > ### Author Response · Authors · 2026-04-08
> > >
> > > Dear Reviewer tzbK,
> > >
> > > We really appreciate your recognition of our work and your kind words, and we are happy to hear that you have a better understanding on your concerns. Again, thank you for your valuable suggestions which have undoubtedly contributed to improving the quality of our paper.
> > >
> > > Many thanks,
> > >
> > > The authors of #27667

---

### Decision · Program_Chairs · 2026-04-30

**Decision:**

Accept (spotlight)

**Comment:**

I concur with the reviewers' very positive view of the submission and recommend acceptance. The paper tackles a natural and important problem, and the formulation, theory, and empirical study were all viewed very positively. The main points for the final version are clearer positioning relative to selective classification, deferral, and CATE or policy-learning alternatives, along with some clarifications around nuisance estimation, calibration details, and rejection-rate control. These are useful refinements, but they do not materially change the overall assessment of a strong contribution.